# Impact of wildfires on particulate matter in the Euro-Mediterranean in 2007: sensitivity to some parameterizations of emissions in air quality models

Marwa Majdi[1-2], Solene Turquety[2], Karine Sartelet[1], Carole Legorgeu[1], Laurent Menut[2], and Youngseob Kim[1]

[1]CEREA: joint laboratory École des Ponts ParisTech – EdF R&D, Université Paris-Est, 77455 Champs sur Marne, France
[2]Laboratoire de Métérologie Dynamique (LMD)-IPSL, Sorbonne Université, CNRS UMR 8539, Ecole Polytechnique, Paris, France.

*Correspondence to:* Marwa Majdi (marwa.majdi@enpc.fr)

**Abstract.**

This study examines the uncertainties on air quality modeling associated with the integration of wildfire emissions in chemistry-transport models (CTMs). To do so, aerosol concentrations during the summer 2007, which was marked by se-
vere fire episodes in the Euro-Mediterranean region especially in Balkan (20–31 July 2007, 24-30 August 2007) and Greece (24-30 August 2007), are analysed. Through comparisons to observations from surface networks and satellite remote sensing, we evaluate the abilities of two CTMs, Polyphemus/Polair3D and CHIMERE, to simulate the impact of fires on the regional particulate matter (PM) concentrations and optical properties. During the two main fire events, fire emissions may contribute up to 90% of surface $PM_{2.5}$ concentrations in the fire regions (Balkans and Greece), with a significant regional impact associated
with long-range transport. Good general performances of the models and a clear improvement of $PM_{2.5}$ and aerosol optical depth (AOD) are shown when fires are taken into account in the models with high correlation coefficients.

Two sources of uncertainties are specifically analysed in terms of surface $PM_{2.5}$ concentrations and AOD using sensitivity simulations: secondary organic aerosol (SOA) formation from intermediate and semi-volatile organic compounds (I/S-VOCs) and emissions' injection heights. The analysis highlights that surface $PM_{2.5}$ concentrations are highly sensitive to injection
heights (with a sensitivity that can be as high as 50% compared to the sensitivity to I/S-VOCs emissions which is lower than 30%). However, AOD which is vertically integrated is less sensitive to the injection heights (mostly below 20%), but highly sensitive to I/S-VOCs emissions (with sensitivity that can be as high as 40%). The maximum statistical dispersion, which quantifies uncertainties related to fire emissions modeling, is up to 75% for $PM_{2.5}$ in Balkan and Greece, and varies between 36 and 45% for AOD above fire regions.
The simulated number of daily exceedance of World Health Organization (WHO) recommendations for $PM_{2.5}$ over the considered region reaches 30 days in regions affected by fires and $\sim 10$ days in fire plumes, which is slightly underestimated compared to available observations. The maximum statistical dispersion ($\sigma$) on this indicator is also large (with $\sigma$ reaching 15 days), showing the need for better understanding of the transport and evolution of fire plumes in addition to fire emissions.

# 1 Introduction

The Mediterranean area is directly affected by large aerosol sources leading to an European maximum in aerosol loading (Putaud et al., 2010; Nabat et al., 2013; Rea et al., 2015). Observations show the influence of a complex mixture of different sources (Dall'Osto et al., 2010; Gerasopoulos et al., 2011; Boselli et al., 2012). The pollution transport pathways in the region are controlled by the very specific orography of this closed sea, but also by the influence of the large circulation patterns (Lelieveld et al., 2002; Lionello et al., 2006) due to its location between the subtropical high-pressure systems, the mid-latitude westerlies and low pressure systems. Mineral dust contributes significantly to pollution episodes in the Euro-Mediterranean area, further increasing aerosol loads associated with local anthropogenic sources (Querol et al., 2009; Gobbi et al., 2007; Kaskaoutis et al., 2008; Nabat et al., 2013; Rea et al., 2015). During summer, high concentrations of organic aerosols, mostly of biogenic origin, are observed in the western Mediterranean (El Haddad et al., 2013; Chrit et al., 2017). Although the precursors are biogenic volatile organic compounds, the formation of organic aerosols is partly explained by the influence of anthropogenic emissions (Kanakidou and Tsigaridis, 2000; Carlton et al., 2010; Sartelet et al., 2012).

Vegetation fires are another sizable sporadic source that needs to be accounted for, especially during summer when the hydrological and meteorological conditions favor their occurrence and spread. Depending on the severity of the fire season, their contribution to the atmospheric aerosol loading and thus to the impairment of the local and regional air quality can be significant (Barnaba et al., 2011; Rea et al., 2015). However, quantifying their contribution remains a challenge due to large uncertainty in emissions and transport.

Most fire episodes in Europe occur in southern countries (Portugal, Spain, France, Italy, Greece) with $\sim 500 \ 10^3$ ha burned every year (Barbosa et al., 2009; Turquety et al., 2014). On average, only $\sim 2\%$ of fires contribute to $\sim 80\%$ of the area burned due to clusters of fires that merge into "mega-fires" (San-Miguel Ayanz et al., 2013). Although ignitions are mainly of anthropogenic origin (negligeance, arson, agricultural practices) according to San-Miguel Ayanz et al. (2013) and the European Forest Fire Information System (EFFIS) of the European Joint Research Center (JRC) (JRC, 2008), fire spread depends on meteorological conditions. It is favored by hot and dry conditions (heat waves and associated droughts), especially if they are preceded by wet winter and spring (fuel accumulation) (Pereira et al., 2005; Hernandez et al., 2015). Fires are also important in eastern Europe, Ukraine, western Russia and Turkey. They are usually smaller and associated with agricultural practices like waste burning and land clearing (Korontzi et al., 2006; Stohl et al., 2007; Turquety et al., 2014).

Turquety et al. (2014) estimate that, on average for the 2003-2010 time period and the Euro-Mediterranean region, total yearly fire emissions amount to $\sim 30\%$ of anthropogenic emissions for $PM_{2.5}$ (particulate matter with diameter $\leq 2.5 \ \mu m$). It is all the more critical as the fire episodes are concentrated during the summer and usually last less than 10 days (so that emissions are very concentrated in time resulting in dense plumes). However, the uncertainty associated with fire emissions is also very large, estimated to $\sim 100$ to $200\%$ (e.g. Urbanski et al. 2011; Turquety et al. 2014). Uncertainties are also linked to the modeling of the temporal variability of emissions. Improving the diurnal cycle may for instance be critical for some fire events (Rea et al., 2016). In addition to emissions, the modeling of wildfires' impact on atmospheric chemistry using chemistry-transport models (CTMs) requires a good knowledge of the emissions' injection height. Indeed, the energy released by fires

can trigger or enhance convection (pyroconvection) and there by injecting emissions at high altitude. Several parameterizations have been developed in recent years (Freitas et al., 2007; Rio et al., 2010; Sofiev et al., 2012) and are increasingly implemented in CTMs. However, comparisons to observations of fire plume's height highlight the difficulty of correctly capturing the vertical shape of fire emissions (Sofiev et al., 2012; Val Martin et al., 2012; Rémy et al., 2016). This will then influence the simulated transport pathways and its regional impact. Injecting above the boundary layer will lower the local impact but result in larger-scale transport. The chemical evolution of fire plumes is still not well understood and not well represented. Primary organic aerosols (POA) are directly emitted by biomass burning into the atmosphere. However, secondary organic aerosols (SOA) are produced through gas-to-particle of oxidation products of volatile organic compounds (VOCs) (with saturation concentration $C^*$ higher than $3.2 \times 10^6 \mu\mathrm{g.m}^{-3}$), intermediate organic compounds (I-VOCs) (with saturation concentration $C^*$ in the range of $320 - 3.2 \times 10^6 \mu\mathrm{g.m}^{-3}$) and semi-volatile organic compounds (S-VOCs) (with saturation concentration $C^*$ in the range $0.32 - 320 \mu\mathrm{g.m}^{-3}$), low volatility organic compounds (L-VOCs) (with saturation concentration $C^*$ lower than $0.32 \mu\mathrm{g.m}^{-3}$) (Lipsky and Robinson, 2006; Grieshop et al., 2009; Huffman et al., 2009; Cappa and Jimenez, 2010; Tsimpidi et al., 2010; Fountoukis et al., 2014; Murphy et al., 2014; Woody et al., 2016; Ciarelli et al., 2017). The formation of secondary organic aerosols (SOA) from the ageing of biomass burning organic precursors is likely to strongly affect aerosol loading and properties in biomass burning plumes. The major organic precursors are thought to be intermediate, semi and low volatility organic compounds (I/S/L-VOCs), (May et al., 2013a; Koo et al., 2014; Konovalov et al., 2015; Ciarelli et al., 2017). However, I/S/L-VOC emissions are not well characterised and their gas-phase emission is often missing from emission inventories (Robinson et al., 2007). Their emissions are often estimated from particulate matter emissions (Couvidat et al., 2012; Ciarelli et al., 2017). The chamber experiments of May et al. (2013a) characterised the volatility distribution of I/S/L-VOCs emissions into different volatility classes. Recent studies have proven that considering I/S-VOCs emitted from biomass burning, shows a major improvement of the agreement between the simulated and observed organic aerosol (OA) (Koo et al., 2014; Konovalov et al., 2015; Ciarelli et al., 2017). Konovalov et al. (2015) find that ignoring I/S-VOCs from biomass burning underestimates strongly the ratio of $\Delta PM_{10} /\Delta CO$ (by a factor of 2) in the city of Kuopio (Finland) and thus leads to an underestimation of the OA concentrations.

The objective of this study is to evaluate the capabilities of two CTMs to simulate the impact of wildfires on the regional particulate matter budget . In the Mediterranean region, surface $PM_{10}$ is dominated by the contribution from dust (Rea et al., 2015). Since the focus of this study is on biomass burning, the discussion is centered on the simulation of surface $PM_{2.5}$. The total loading of aerosols over the region is evaluated using comparisons of AOD to observations. After an evaluation of two CTMs (CHIMERE and Polyphemus/Polair3D) through comparisons to observations from surface networks and remote sensing, a sensitivity analysis to the injection heights and I/S-VOCs emissions is conducted in order to quantify the uncertainties associated to these two parameterizations, in terms of AOD and surface PM concentrations. This analysis is focused on the summer 2007 which was marked by extreme meteorological conditions (consecutive heat waves and drought) and severe fire episodes in the eastern Mediterranean and Europe. According to the European Forest Fire Information System (EFFIS), 2007 was well above the average of the previous last 3 decades in terms of burned area (574 361 against 495 471 ha burned respectively) (JRC, 2008). The burning season was particularly severe for Greece where the burned area reached extreme values.

This case study is particularly interesting since it was well captured by satellite sensors as the resulting smoke plumes, fanned by north-easterly winds, were transported over the sea crossing the south Ionian Sea and reaching the northern part of the African continent. Several studies have highlighted the important enhancements in atmospheric gases (Turquety et al., 2009; Coheur et al., 2009; Hodnebrog et al., 2012) and aerosols (Liu et al., 2009; Kaskaoutis et al., 2011) due to these fire episodes. Modeling analyses have investigated secondary production in the fire plumes. While Hodnebrog et al. (2012) showed limited ozone impact on average during the summer, Poukpou et al. (2014) investigated more precisely the fire event in Peloponnese (Greece) at the end of August 2007. They found enhancements of CO and $NO_x$ concentrations mainly over the burnt areas due to the biomass burning. Due to the non-linear dependence of $O_3$ on $NO_x$ levels, the near surface $O_3$ values were reduced (-34%) over the Poleponnese, but increased (+52%) over the sea at 500 km downwind. Here, the sensitivity of regional aerosol loadings, both primary and secondary, to modeling configurations are analyzed.

This paper is structured as follows. In the second section, a brief description of the chemistry transport models and the different tools and methodology used in this work, are given. Then, section 3 presents statistics for model-to-data comparison to assess the models performance during the summer 2007 and more specifically during the main fire events. Then, uncertainties related to the integration of wildfires in CTMs (injection height, I/S-VOCs emissions) are discussed in section 4. Finally, section 5 describes the contribution of wildfires to air quality threshold exceedences as well as the associated uncertainties.

## 2 Simulation experiments

### 2.1 Chemistry-transport models

Two CTMs are used for this study. CHIMERE simulations (Menut et al., 2013) allow us to perform inter-model comparison and to evaluate the capability of other current CTMs to simulate the impact of wildfires on the regional particulate matter budget and to quantify the uncertainties on air quality modeling related to the integration of fire emissions in CTMs. The sensitivity analysis is undertaken using the Polyphemus modeling platform of air quality (Mallet et al., 2007) with the chemistry transport model Polair3D (Sartelet et al., 2012).

Table 2.1 summarizes the main characteristics of the simulations.

**Table 2.1.** Main characteristics of the Polyphemus and CHIMERE simulations.

| Meteorology | European Center for Medium-Range Weather Forecasts (ECMWF, ERA-Interim) model |
|---|---|
| Boundary conditions | From nesting simulation: large domain ($0.5° \times 0.5°$, horizontal resolution) covering Europe and North Africa (Figure A1 in Appendix A) |
| Chemical mechanism | - Polyphemus: Carbon Bond 05 model (CB05) (Yarwood et al., 2005) for the gas-phase chemistry (modified following (Kim et al., 2011) for SOA formation) <br> - CHIMERE: Modele lagrangien de chimie de l'ozone a l'echelle Regionale 2 (Melchior-2) (Derognat et al., 2003) |
| Horizontal resolution | - Large domain: $0.5° \times 0.5°$ <br> - Small domain: $0.25° \times 0.25°$ |
| Vertical resolution | - Polyphemus: 14 levels (surface–12km) <br> - CHIMERE: 19 levels (surface–200hPa) |
| Biogenic emissions | Model of Emissions of Gases and Aerosols from Nature (MEGAN) (Guenther et al., 2006): <br> - Polyphemus: the standard MEGAN LHIV database MEGAN-L <br> - CHIMERE: MEGAN v2.04 |
| Anthropogenic emissions | EMEP emissions inventory for 2007 (European Monitoring and Evaluation Program,www.emep.int) |
| Fire emissions | APIFLAME fire emissions' model v1.0 described in Turquety et al. (2014) |
| Dust emissions | Surface and soil databases (Menut et al., 2013) <br> Briant et al. (2017) |
| Sea-salt emissions | Parameterization of (Monahan, 1986) |

For the simulations presented in this work, Polyphemus is used with the Carbon Bond 05 model (CB05) (Yarwood et al., 2005) for the gas-phase chemistry (modified following Kim et al. (2011) for SOA formation) and with the SIze resolved Aerosol Model (SIREAM) (Debry et al., 2007) for aerosol dynamics (coagulation, condensation/evaporation). SIREAM uses a sectional approach and the dry particle diameter is discretized with 5 sections between 0.01 μm and 10 μm. The simulations are performed on 14 vertical levels extending from the ground to 12 km.

The version 2016 of the regional CTM CHIMERE is used for this work (Menut et al., 2013; Mailler et al., 2016). Simulations were conducted using the reduced chemical mechanism Modele Lagrangien de Chimie de l'Ozone a l'echelle Regionale 2 (MELCHIOR2), which includes 44 species and 120 reactions, and the aerosol module described in Bessagnet et al. (2004)

(nucleation, coagulation, absorption). This module is also based on a sectional representation of the size distribution. For this simulations, 10 bins from 40 nm to 40 μm are used and simulations are performed on 19 vertical levels extending from the surface to 250 hPa ($\sigma$-pressure coordinates).

Both models include wet and dry deposition. Deposition in Polyphemus is described in Sartelet et al. (2007) and in CHIMERE in Menut et al. (2013) and Mailler et al. (2017). Thermodynamics of inorganic aerosols are modeled using ISORROPIA (Nenes et al., 1999) with a bulk equilibrium approach. Bulk equilibrium is also used for SOA formation, and the partitioning between the gas and particle phases is done with The Secondary Organic Aerosol Processor (SOAP) (Couvidat and Sartelet, 2015) in Polyphemus. Photolysis rates are calculated using the FastJX model (version7.0b for CHIMERE) (Wild et al., 2000). Their
online calculation in CHIMERE allows to represent the attenuation by clouds and aerosols, while the attenuation by clouds in Polyphemus is modeled by multiplying the clear-sky photolysis rates by a correction factor (Real and Sartelet, 2011).

Both models (Polyphemus and CHIMERE) are driven by meteorological conditions simulated by the European Center for Medium-Range Weather Forecasts (ECMWF, ERA-Interim) model. Initial and boundary conditions from MOZART4-GEOS5 6-hourly simulation outputs are used (Emmons et al., 2010). Simulations are undertaken using two nested domains (Figure A1
in Appendix A). One large domain ($0.5° \times 0.5°$ horizontal resolution) covering Europe and North Africa to provide realistic dust sources and a smaller domain at $0.25° \times 0.25°$ horizontal resolution over the Mediterranean area (presented in Figure 2.1).

Anthropogenic emissions are derived from the EMEP emissions inventory for 2007 (European Monitoring and Evaluation Program, www.emep.int). The inventory species are disaggregated into real species using speciation coefficients (Passant, 2002). The aggregation into model species is done following Middleton et al. (1990). Primary particulate matter emissions are
given in total mass by the EMEP emission inventory. They are speciated into dust, primary organic emissions (POA) and black carbon (BC), and distributed into 5 diameter bins (Sartelet et al., 2007).

Biogenic emissions of isoprene and terpenes ($\alpha$-pinene, $\beta$-pinene, limonene and humulene) are calculated using the Model of Emissions of Gases and Aerosols from Nature with the standard MEGAN LHIV database MEGAN-L for Polyphemus and MEGAN v2.04 for CHIMERE (Guenther et al., 2006). Sea-salt emissions are parameterized following Monahan (1986). The mineral dust emissions are calculated using soil and surface databases (Menut et al., 2013) and with a spatial extension of potentially emitting area in Europe as described in Briant et al. (2017). The daily fire emissions are detailed in section 2.2.

In Polyphemus, I/S/L-VOC emissions are estimated by multiplying the primary organic emissions (POA) by a factor of 1.5, following the chamber measurements (Robinson et al., 2007; Zhu et al., 2016; Kim et al., 2016). The factor of 1.5 is used for
both anthropogenic and fire emissions to estimate the gas-phase I/S/L-VOCs that are not included in the inventories. I/S/L-VOCs emissions are assigned to 3 surrogates species: POAlP, POAmP and POAhP (for compounds of low, medium and high volatilities respectively), of saturation concentration $C^*$: $\log_{10}(C^*)$ = -0.04, 1.93, 3.5 respectively. The volatility distribution at emissions of I/S/L-VOCs is detailed in Couvidat et al. (2012) (25%, 32%, and 43% of I/S/L-VOCs are assigned to POAlP, POAmP and POAhP respectively). It corresponds to the volatility distribution measured by May et al. (2013a) for biomass
burning aerosol emissions. Each primary I/S/L-VOCs undergoes one OH-oxidation reaction in the gas phase with a kinetic rate constant equal to $2.10^{-11}$ molecule$^{-1}$.cm$^3$.s$^{-1}$, leading to the formation of secondary surrogates: SOAlP, SOAmP and SOAhP. The ageing of the primary aerosols reduces the volatility of the secondary product by a factor of 100 and increases the

molecular weight by 40% (Couvidat et al., 2012).

## 2.2 Fire emissions

Daily fire emissions are calculated using the APIFLAME fire emissions' model v1.0 described in Turquety et al. (2014). The carbon emission associated with a specific fire is calculated using the MODIS burned area product at 500 m resolution (MCD64 product) (Giglio et al., 2009), multiplied by the consumed fuel load specific to the vegetation burned. The CORINE LAND COVER (CLC) is used for vegetation attribution, and the biomass density is estimated based on simulations by the ORCHIDEE vegetation dynamics and carbon cycle model (Maignan et al., 2009). Turquety et al. (2014) estimated an uncertainty of ∼100% on daily carbon emissions using an ensemble approach. This is in agreement with estimates for other daily inventories (e.g. GFED (Van der Werf et al., 2010)).

Emissions for each species are derived from the carbon emissions using the emissions factors from Akagi et al. (2011). These emission factors are provided in terms of g species per Kg Dry Matter (DM) burned ($g.kg^{-1}$) for all relevant species observed in biomass burning plumes and for different standard vegetation types that match to Mediterranean landscapes (chaparral, temperate forest, crop residue, pasture maintenance and savanna). The contribution of these vegetation types to the burned area detection over the Mediterranean region during the time period studied is 37.2% for temperate forest, 32.7% for savanna, 9.6% for chaparral and 19.9% for crop residue. Emissions for inventory species are then converted into emissions for model species using model-specific aggregation matrices (Yarwood et al., 2005). For aerosols, the difference between emissions factors provided for the total $PM_{2.5}$ and for the main primary emissions (organic and black carbon, small amounts of inorganics) is modelled as a specific, inert and unidentified species grouping the remaining mass of $PM_{2.5}$ (REM-$PM_{2.5}$). REM-$PM_{2.5}$ corresponds to all the unidentified fine particles emitted by wildfires which are incorporated to consider the differences between $PM_{2.5}$ emissions and the total of all $PM_{2.5}$ speciated emissions. The difference between $PM_{10}$ and $PM_{2.5}$ emission factors is attributed to emissions of coarse mode PM ($PPM_{coarse}$).

Figure 2.1 shows a map of total primary organic carbon emissions (OC) from fires in the Euro-Mediterranean region during the summer 2007. Four main areas are affected by wildfires: Balkan (sub-region MedReg1), Greece (sub-region MedReg2), Southern Italy (sub-region MedReg3) and Algeria (sub-region MedReg4).

Total daily fire emissions for the four studied areas are plotted in Figure 2.2. In all regions, fire emissions are occasional but very intense. The largest fires in the simulated domain occur in Balkan (sub-region MedReg1) between 20 July and 31 July 2007, and in Greece (sub-region MedReg2) between 24 August and 30 August. In Algeria (sub-region MedReg4), fires mainly occur at the end of August and beginning of September (28 August – 1 September). In Southern Italy (sub-region MedReg3), fires are observed between 9 July and 31 July 2007.

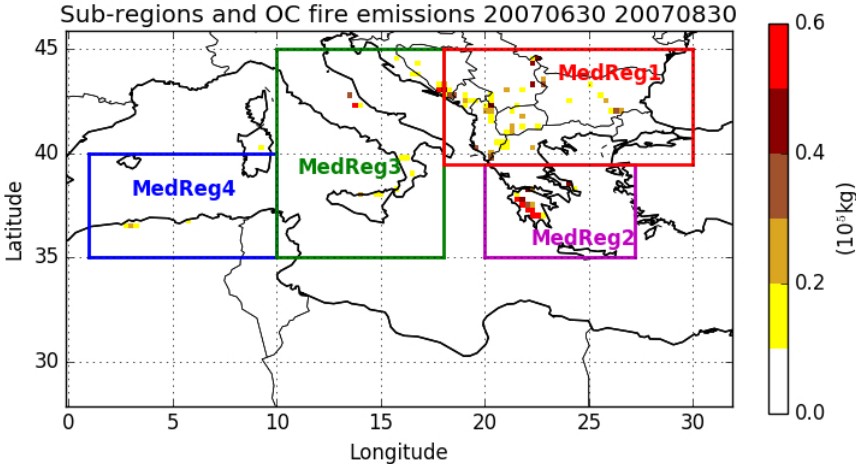

**Figure 2.1.** Map of the nested domain over the Mediterranean area with a spatial resolution of $0.25° \times 0.25°$. The total organic carbon emissions (kg. (grid cell)$^{-1}$) from fires during the summer of 2007 are presented. The sub-regions used in this study are also indicated in colored boxes: MedReg1 (Balkan + Eastern Europe), MedReg2 (Greece), MedReg3 (Italy) and MedReg4 (Algeria).

## 2.3 Model sensitivity experiments

Two different parameters, considered critical for modeling fire events, are tested using sensitivity simulations with the Polyphemus CTM: fire emissions of I/S-VOCs and emissions' injection heights. Another simulation is also conducted to evaluate the impact of REM-PM$_{2.5}$ emissions. In the simulations discussed here, four different configurations of the model are used:

1. Simulation *Poly-ref* : fire emissions are homogeneously mixed the planetary boundary layer (PBL), but no lower than 1 km. The percentage of fire emissions injection is divided homogeneously depending on layers' thickness (I/S/L-VOCs included).

2. Simulation *Poly-noI/S-VOCs* : fire emissions in the PBL but without I/S-VOC emissions in the gas phase. In this sensitivity study, the semi-volatile properties of POA are not considered, and POA emissions are modeled as LVOCs.

3. Simulation *Poly-NoREM-PM$_{2.5}$* : fire emissions in the PBL, with I/S/L-VOCs but without REM-PM$_{2.5}$.

4. Simulation *Poly-3km* : fire emissions injected up to 3km with: 20% under 1km, 80% between 1 and 3km. Note that, in this case, 78% of fire emissions are injected above the PBL. This choice of sensitivity study may be viewed as conservative since, for example, injection heights are limited to 3 km. But it is also extreme since maximum injection at 3 km is imposed to all fires, resulting in injection above the boundary layer. This could be realistic, since based on the Multi-angle Imaging SpectroRadiometer (MISR) observations, Mims et al. (2010) estimated that 26% of the fire plumes exceed the boundary layer.

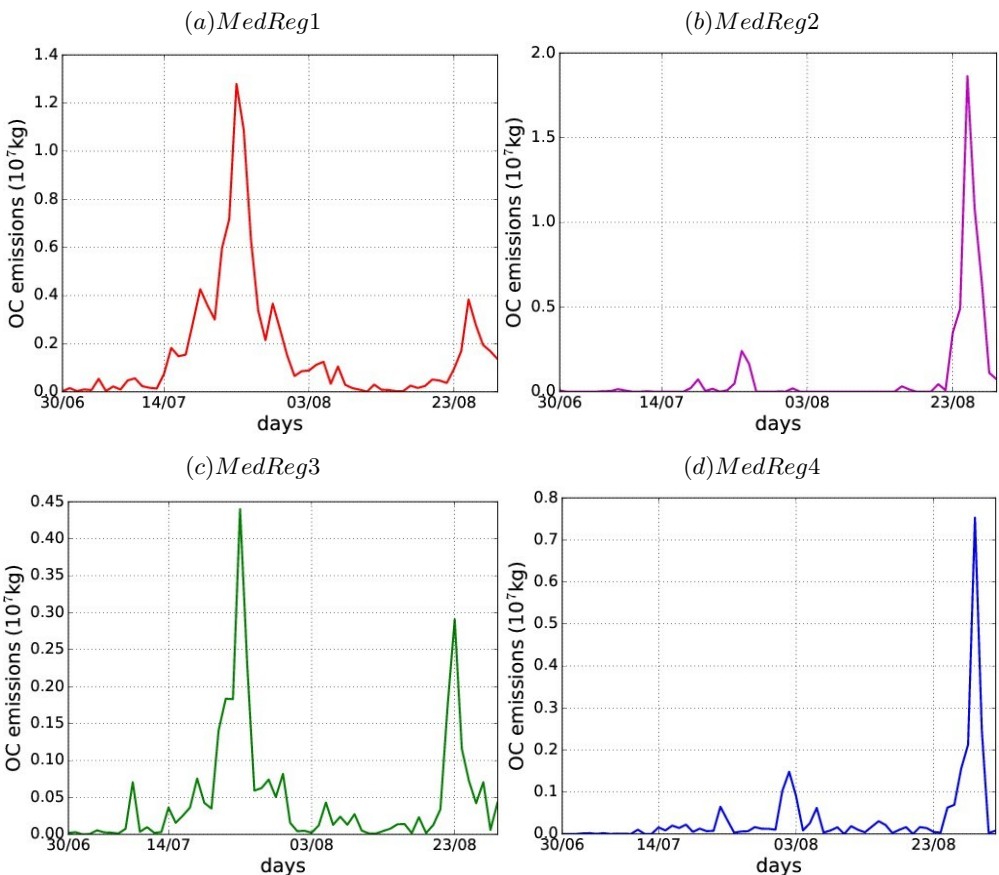

**Figure 2.2.** Daily total OC emissions calculated by APIFLAME during the summer of 2007 in the four sub-regions of Figure 2.1.

In addition, a simulation with the CHIMERE model (without I/S-VOCs and with fires in the PBL)(*CHIMERE-ref*) allows an inter-model comparison. Simulations without fires are also computed with both models (*Poly-Nofires* and *CHIMERE-Nofires*). The set up of the different simulations is summarized in Table 2.2.

**Table 2.2.** Summary of the configurations used in different simulations. (N/A: not applicable)

| Simulation | Fire emissions | I/S-VOCs from fire | Fire emissions'injection height | REM-PM$_{2.5}$ |
|---|---|---|---|---|
| Simulation *Poly-ref* | Yes | Yes | Between 1km and PBL | Yes |
| Simulation *Poly-3km* | Yes | Yes | 20% under 1km<br>80% between 1 and 3km | Yes |
| Simulation *Poly-NoREM-PM$_{2.5}$* | Yes | Yes | Between 1km and PBL | No |
| Simulation *Poly-noI/S-VOCs* | Yes | No | Between 1km and PBL | Yes |
| Simulation *CHIMERE-ref* | Yes | No | Between 1km and PBL | Yes |
| Simulation *CHIMERE-Nofires* | No | N/A | N/A | N/A |
| Simulation *Poly-Nofires* | No | N/A | N/A | N/A |

# 3 Model evaluation

## 3.1 Observations

Daily observations from the European network AIRBASE 5 are used for PM concentrations. Due to the relatively coarse
horizontal resolution, only background stations are included in the present paper. The investigated stations are indicated in
Figure 3.1. PM$_{10}$ concentrations in the Mediterranean area are strongly affected by dust, which are difficult to simulate due
to their sporadic nature and the fact that their main sources are located out of the model domain. Although dust emission is
modeled with state-of-the-art parameterization in this study, the analysis focuses on PM$_{2.5}$ in order to evaluate more specifically
the uncertainties associated to wildfires and to minimize the contribution from dust.

Surface observations are complemented by remote sensing observations of aerosol optical properties. AOD level 2.0 data
(at 550 nm) and Angstrom coefficient $\alpha$ (derived from AOD at 500 and 870 nm) from the AERONET (AErosol RObotic
NETwork) ground-based network of sun photometers (Holben et al., 1998) are used. The uncertainty on AOD is estimated to
less than 0.02 (Holben et al., 2001). AOD level 2.0 observations are missing in Lecce University (in Italy) during the first event,
and in Blida (in Algeria) during the second one. Since AOD Level 1.5 observations at 500 nm (before filtering) are available
for the latter, we have chosen to include these observations for comparison in the next sections. The considered stations are
indicated in Figure 3.1.

AOD observations at 550 nm from the MODIS instrument on board the Terra (equator crossing time at 10:30, ascending
node) and Aqua (equator crossing time at 13:30, ascending node) satellites are also used in order to get a more complete
regional view. The MOD04 and MYD04 (for Terra and Aqua, respectively) from collection 5.2 data, available at $10 \times 10$ km$^2$
are used (Remer et al., 2005). The expected uncertainty on AOD is $\Delta\tau = \pm 0.05 \pm 0.15\tau$ ($\tau$ is the optical thickness) over land
(Chu et al., 1998; King et al., 1999) and $\Delta\tau = \pm 0.03 \pm 0.05\tau$ over ocean (King et al., 1999), with a good agreement with ground
based measurements (Remer et al., 2005). Deep Blue AOD (Sayer et al., 2013) is used when available (over bright areas).

### 3.2 Comparison method

A set of statistical indicators are used for the comparison of model simulations to surface measurements: the root mean square error ($RMSE$), the correlation coefficient ($R$), the mean fractional bias ($MFB$) and the mean fractional error ($MFE$).

These are defined as:

$$\text{RMSE} = \sqrt{\frac{1}{n} \sum_{i=1}^{n} (c_i - o_i)^2} \tag{1}$$

$$\text{R} = \frac{\sum_{i=1}^{n} (c_i - \bar{c})(o_i - \bar{o})}{\sqrt{\sum_{i=1}^{n} (c_i - \bar{c})^2} \sqrt{\sum_{i=1}^{n} (o_i - \bar{o})^2}} \tag{2}$$

$$\text{MFB} = \frac{1}{n} \sum_{i=1}^{n} \frac{c_i - o_i}{(c_i + o_i)/2} \tag{3}$$

$$\text{MFE} = \frac{1}{n} \sum_{i=1}^{n} \frac{\mid c_i - o_i \mid}{(c_i + o_i)/2} \tag{4}$$

with $o_i$ the observed concentration at time and location $i$, $c_i$ the modeled concentration at time and location $i$, and $n$ the number of data.

Boylan and Russel (2006) proposed for PM that a model performance criterion (level of accuracy acceptable for standard modeling applications) is met when $MFE \leq +75\%$ and $MFB$ is within $\pm$ 60%, and a model performance goal (level of accuracy considered to be close to the best a model can be expected to achieve) is met when $MFE \leq 50\%$ and $MFB$ is within $\pm30\%$. In the following, the $MFB$ and $MFE$ are computed at each station and averaged.

### 3.3 Overview of the three months period

The statistical evaluation of the simulations during the summer 2007 (from 15 July to 30 August 2007) is presented in Table 3.1 for PM$_{2.5}$ concentrations and Table 3.2 for AOD at 550 nm. Globally, the PM$_{2.5}$ concentrations and AOD are well reproduced by the models, although they are slightly underestimated compared to measurements. For AOD, the model performance and goal are always met. For PM$_{2.5}$ concentrations, the model performance is always met, and the model goal is

10 met for the model errors MFEs. The model-to-measurements correlations range between 46% and 83% for all the simulations when fires are included. The model errors MFEs are similar for outputs from CHIMERE and Polyphemus. However, CHIMERE outputs have lower biais (higher concentrations are AOD values), while Polyphemus outputs have higher correlations. The models-to-measurements comparisons tend to improve when fire are taken into account, with lower MFEs and higher correlations. The improvement is stronger for AOD than for PM$_{2.5}$ concentrations, because the stations used for AOD models-to-measurements comparisons are closer to regions affected by wildfires than the stations used for PM$_{2.5}$ models-to-measurements comparisons, as discussed below.

Figure 3.1 shows the mean surface concentrations of PM$_{2.5}$ simulated by Polyphemus (*Poly-ref*) over the Euro-Mediterranean domain, from 15 July to 30 August 2007. The 8 AIRBASE stations used for the models-to-measurements comparison are also presented in Figure 3.1. The mean simulated PM$_{2.5}$ concentrations can reach 60 $\mu$g.m$^{-3}$ in regions affected by wildfires (Balkan, sub-region MedReg1 and Greece, sub-region MedReg2). Only two AIRBASE stations, GR0039A in Greece and IT0459A, in Italy are close to regions affected by wildfires.

Figure B1 in Appendix B shows that conclusions for CHIMERE are similar to those for Polyphemus.

Figure 3.1 also shows the mean modeled AOD, which can be as high as 0.72 in average in Balkan and in Greece, and the 6 Aeronet stations used for the models-to-measurements comparisons. 5 out of the 6 stations are located in regions affected by wildfires.

This evaluation shows good general performance of the models and a clear improvement of PM$_{2.5}$ and AOD when fires are included, allowing a more precise analysis of the model's behavior during the strongest fire events.

**Table 3.1.** Statistics of models-to-measurements comparisons for the mean daily PM$_{2.5}$ concentrations during the summer 2007 (AIRBASE station number = 8).

| Simulations | Mean observed PM$_{2.5}$ | Mean simulated PM$_{2.5}$ | Correlation (%) | MFB (%) | MFE (%) |
|---|---|---|---|---|---|
| *Poly-ref* | 13.2 | 9.3 | 82 | -32 | 42 |
| *Poly-noI/S-VOCs* | 13.2 | 9.3 | 82 | -32 | 42 |
| *Poly-3km* | 13.2 | 9.1 | 82 | -33 | 43 |
| *Poly-NoREM-PM$_{2.5}$* | 13.2 | 9.1 | 82 | -33 | 43 |
| *Poly-Nofires* | 13.2 | 8.4 | 78 | -37 | 46 |
| *CHIMERE-Nofires* | 13.2 | 11.2 | 70 | -15 | 39 |
| *CHIMERE-ref* | 13.2 | 11.3 | 67 | -10 | 39 |

**Table 3.2.** Statistics of models-to-measurements comparisons for mean daily AOD at 550 nm during the summer 2007 (AERONET station number = 6).

| Simulations | Mean observed AOD | Mean simulated AOD | Correlation (%) | MFB (%) | MFE (%) |
|---|---|---|---|---|---|
| *Poly-ref* | 0.27 | 0.22 | 62 | -14 | 34 |
| *Poly-noI/S-VOCs* | 0.27 | 0.21 | 64 | -18 | 35 |
| *Poly-3km* | 0.27 | 0.22 | 64 | -15 | 34 |
| *Poly-NoREM-PM$_{2.5}$* | 0.27 | 0.22 | 62 | -14 | 34 |
| *Poly-Nofires* | 0.27 | 0.19 | 56 | -24 | 39 |
| *CHIMERE-Nofires* | 0.27 | 0.23 | 36 | -7 | 39 |
| *CHIMERE-ref* | 0.27 | 0.24 | 46 | -6 | 36 |

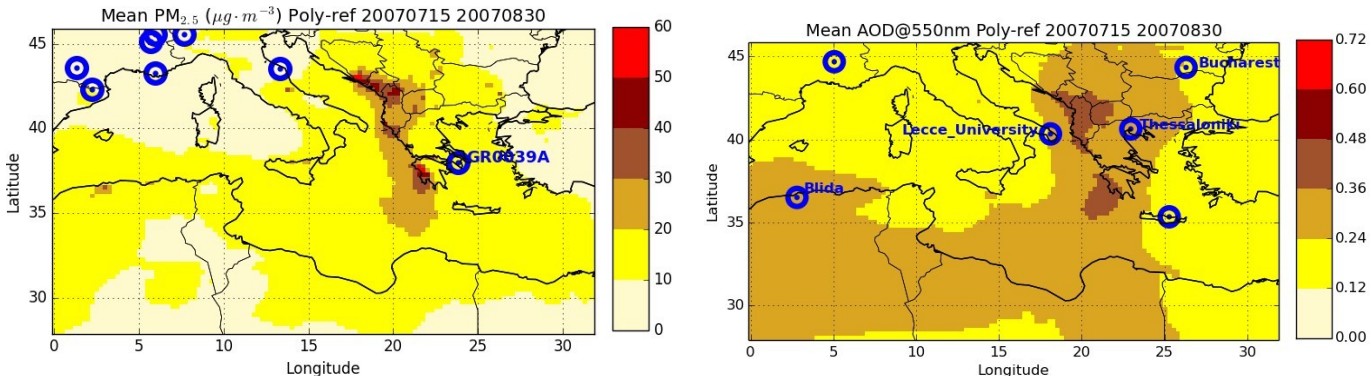

**Figure 3.1.** Daily mean surface PM$_{2.5}$ and AOD at 550 nm from the *Poly-ref* simulation averaged over the summer of 2007 (the 8 AIRBASE and 6 AERONET stations, used in this work, are represented here in blue dots).

## 3.4 Fire events

To better understand the sensitivity of PM$_{2.5}$ concentrations and AOD to I/S-VOCs emissions and the fire injection heights during the fire events, the composition of PM during fire peaks and the evaluation of fire contribution are examined. The locations of the fire peaks during the two main fire events are first detailed.

### 3.4.1 Locations of the fire peaks

The contribution of fires to PM$_{2.5}$ concentrations simulated by Polyphemus during the two main fire events (20-31 July and 24-30 August) is presented in Figure 3.2 as the relative difference between the reference simulation (*Poly-ref*) and the simulation without fire emissions (*Poly-Nofires*). The largest contribution is simulated over the Balkan and Eastern Europe during the first period (sub-region MedReg1), and Greece (sub-region MedReg2) and Algeria (sub-region MedReg4) during the second

period. The impact reaches up to 90% locally on average during each fire event. The contribution of fires remains large (>60%) over most of the eastern Mediterranean basin, and part of the western basin at the end of August due to long-range transport of fire plumes. Figure C1 in Appendix C shows that the contribution of fire for CHIMERE are similar to that for Polyphemus.

20 Only the closest AIRBASE and AERONET stations to fire regions (where fire contribution is higher than 50%) are used in models-to-measurements comparisons in the next sections. During the first fire event, the most affected stations are: GR0039A in Greece (sub-region MedReg2) and Bucharest in Romania (sub-region MedReg1), while during the second fire event, stations in Greece (Thessaloniki in sub-region MedReg2), Italy (Lecce University in sub-region MedReg3) and in Algeria (Blida in sub-region MedReg4) are the most influenced. However, since the models show the same behavior during the second fire event over the stations GR0039A and Thessaloniki, we choose to focus only on GR0039A.

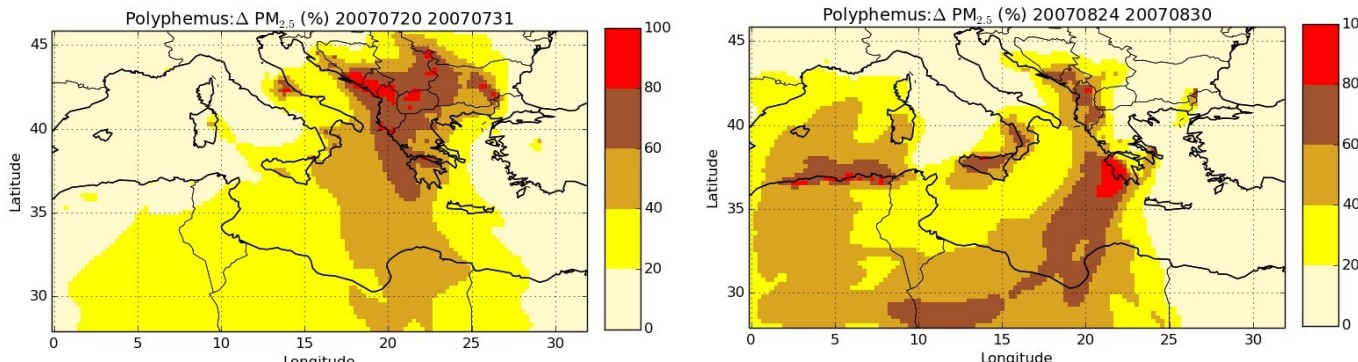

**Figure 3.2.** Left panel: relative difference of surface PM$_{2.5}$ concentrations between simulations *Poly-ref* and *Poly-Nofires* during the first fire event. Right panel: relative difference of surface PM$_{2.5}$ concentrations between simulations *Poly-ref* and *Poly-Nofires* during the second fire event.

### 3.4.2 Aerosol composition during fire peaks

The composition of surface PM$_{2.5}$ concentrations simulated during the first fire event over MedReg1 and during the second one over MedReg2 are shown in Figures 3.3 and 3.4 respectively. These two subregions are the areas most affected by fires (high fire emissions (Figure 2.1) especially during the first fire event for Medreg1 and during the second fire event for MedReg2.

10 The upper left panel shows the composition of surface PM$_{2.5}$ concentrations for the simulation without fire *Poly-Nofires* (background surface PM$_{2.5}$ concentrations), while the upper right panel shows the composition of surface PM$_{2.5}$ concentrations due to fires (differences between the simulations *Poly-ref* and *Poly-Nofires*). If wildfires are not taken into account (simulation without fire) and during the first fire event over MedReg1, organic and inorganic aerosols contribute equally (42.6%, 40.5%) to the surface PM$_{2.5}$ concentrations. The contribution of REM-PM$_{2.5}$ (dust), black carbon are lower (15%, 1%). As noted by Chrit

15 et al. (2017), most of summer organic aerosols are from biogenic sources in this region. If wildfires are not taken into account during the second fire event over MedReg2, inorganics and REM-PM$_{2.5}$ are the predominant component in the composition

of PM$_{2.5}$ concentrations (56.5% and 27.9%). Lower contributions are simulated for black carbon (1.2%) and organic aerosol (14.3%). Figures 3.3 and 3.4 also show the composition of surface PM$_{2.5}$ concentrations due to fires for the simulations *Poly-ref* (differences between the simulations *Poly-ref* and *Poly-Nofires*), *Poly-noI/S-VOCs* (differences between the simulations *Poly-noI/S-VOCs* and *Poly-Nofires*), *Poly-NoREM-PM$_{2.5}$* (differences between the simulations *Poly-NoREM-PM$_{2.5}$* and *Poly-Nofires*). During the first fire event over MedReg1, organic aerosol is predominant in the contribution of fires (between 47% and 85% of the contribution). Organic aerosol is mostly composed of POA and SOA from I/S/L-VOCs (46 to 80%). Note that POA and SOA from L-VOCs (low volatile organic compounds) are important even in the simulation when I/S-VOCs are not taken into account in fire emissions (46%), because POA are then assigned to L-VOCs. The contribution from inorganics (8 to 13%) and black carbon (3 to 6%) are low. During the second fire event over MedReg2, similar PM$_{2.5}$ composition is found. Organic aerosol (mainly the POA and SOA from I/S/L-VOCs) is the most important component contributing to PM$_{2.5}$ from fires (between 46% and 81% of the contribution). The contribution from inorganics (9 to 12%) and black carbon (5 to 6%) are lower. The contribution of REM-PM$_{2.5}$ from fires (if it is included) is very significant, 27% for the simulations with I/S-VOCs and 36% otherwise. Because REM-PM$_{2.5}$ emissions are incorporated to consider the difference between PM$_{2.5}$ emission factors and the total of all PM included in specific species, the contribution of REM-PM$_{2.5}$ may be overestimated (double counting with I/S-VOCs for instance).

In our study, inorganics (mainly sulfate, sea salt and ammonium) contribute highly to PM$_{2.5}$ composition, if fire emissions are not considered. Similar results are found in Fountoukis et al. (2011) who showed the high contribution of sulfate, sea salt and ammonium to PM over Europe during May 2008. However, when fire emissions are taken into account, the contribution of inorganics becomes lower than the contribution of organics (8 to 9% of inorganics against 40 to 80% of organics). Focusing on the contribution from fires, sulfate, ammonium and nitrate are the predominant components in the composition of inorganics from fires: between 55.7% and 67.6% for sulfate, between 26.8 and 38.7% for ammonium and 5.6 to 13.6% for nitrate.

Similar PM$_{2.5}$ composition is found during the second fire peak, and in the concentrations simulated by CHIMERE (not shown here). Surface PM$_{2.5}$ concentrations from fire simulated by CHIMERE are composed in the first and second events mainly of organic aerosols, mostly composed by primary organic carbon (OCAR) (46%) which corresponds to I/S/L-VOCs in Polyphemus, and of REM-PM$_{2.5}$ (39%). The contributions from inorganics (9%), black carbon (5.2%) and SOA (2.7%) are low.

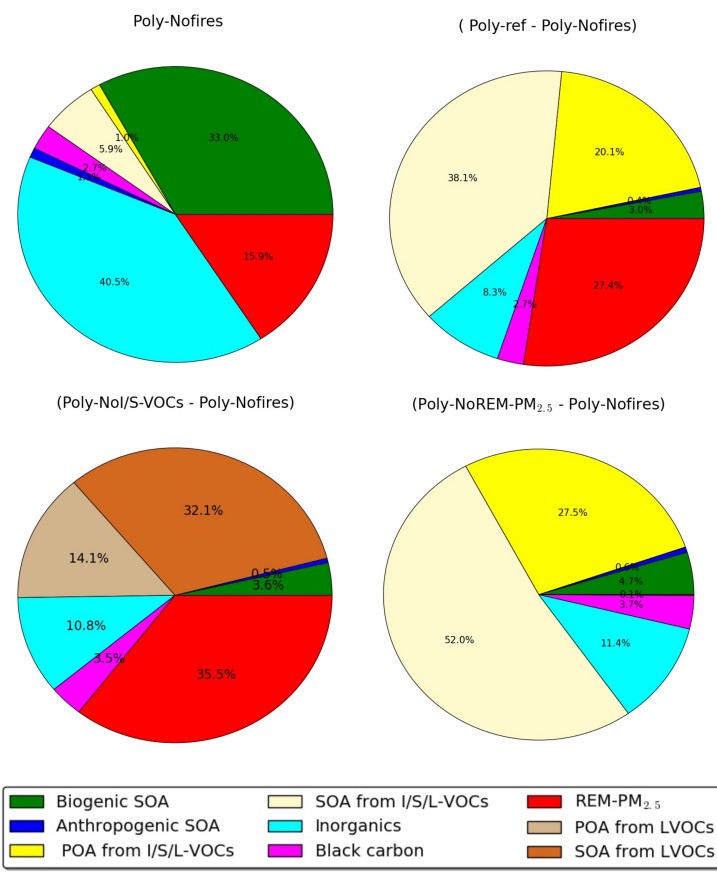

**Figure 3.3.** Composition of surface PM$_{2.5}$ concentration over MedReg1 during the first fire event (upper left panel, simulation Poly-Nofires). Composition of surface PM$_{2.5}$ concentration due to fires (upper right panel: simulation Poly-ref; lower left panel: simulation Poly-noI/S-VOCs; lower right panel: simulation Poly-NoREM-PM$_{2.5}$).

### 3.4.3 Evaluation of fire contribution

10     Figure 3.5 shows time series of daily observed and simulated aerosols at background suburban and background rural stations with available data during fire events and with a fire contribution higher than 10% (PM$_{2.5}$ in Greece (sub-region MedReg2), AOD at 550 nm in Italy (sub-region MedReg3), Romania (sub-region MedReg1) and Algeria (sub-region MedReg4). A significant increase in AOD and PM$_{2.5}$ concentration is observed during the major fire episodes on 20–28 July and 24–30 August, associated with large contributions from fires. The daily average AOD, observed by MODIS and simulated for the sub-regions of Figure 2.1, are shown in Figure 3.6. Background AOD and daily variability are consistent with MODIS for both models in the sub-regions of Figure 2.1, with high correlation coefficients (∼90% on average for Polyphemus, 80% for CHIMERE).

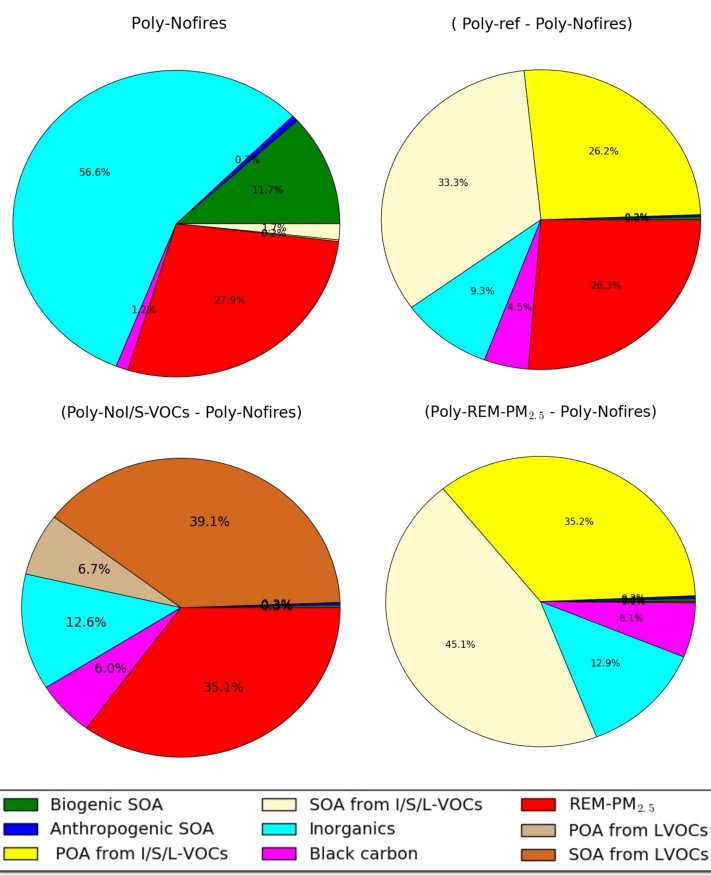

**Figure 3.4.** Composition of surface PM$_{2.5}$ concentration over MedReg2 during the second fire event (upper left panel, simulation Poly-Nofires). Composition of surface PM$_{2.5}$ concentration due to fires (upper right panel: simulation Poly-ref; lower left panel: simulation Poly-noI/S-VOCs; lower right panel: simulation Poly-NoREM-PM$_{2.5}$).

As shown in Figures 3.5 and 3.6, injecting emissions higher (simulation *Poly-3km*) significantly lowers surface PM concentrations (compared to the simulation *Poly-ref*), even if the maximum injection height remains conservative. Not taking into account SOA from I/S-VOCs directly reduces emissions thus having strong influence on PM concentrations (-20% compared to

*Poly-ref*). A reduction of the same order of magnitude is obtained when REM-PM$_{2.5}$ is not accounted for (*Poly-NoREM-PM$_{2.5}$*), suggesting that the incorporation of this species in APIFLAME could compensate for the missing I/S-VOCs emissions.

Compared to MODIS AOD (Figure 3.6), the simulations including I/S-VOCs (*Poly-ref*, *Poly-3km* and *Poly-NoREM-PM$_{2.5}$*) overestimate AOD during the fire events, while simulations without I/S-VOCs underestimate AOD. This is more pronounced in the two sub-regions MedReg1 and MedReg2, where the mean modeled AOD values are overestimated by about 30% for

*Poly-ref* and *Poly-3km* and 10% for *Poly-NoREM-PM$_{2.5}$*. Compared to AIRBASE ground measurements (Figure 3.5), the peak corresponding to the first event (25 July) is well modeled in the simulation *Poly-ref* compared to PM$_{2.5}$ observations

at GR0039A (Athens suburbs, Greece, sub-region MedReg2). The temporal variations of the mean simulated $PM_{2.5}$ concentrations are consistent with observations, with high correlation coefficients (>88%). Background levels of $PM_{2.5}$ are slightly underestimated compared to observations. This can be explained by an underestimation of dust long-range transport or an underestimation of local emissions. The peak of $PM_{2.5}$ concentration is slightly underestimated in all the other simulations. At the GR0035A station, the temporal tendencies of the simulated $PM_{2.5}$ concentrations are consistent with the observations. However, the first $PM_{2.5}$ peak is underestimated since $PM2_{2.5}$ levels are strongly underestimated (bias= -85%). The observed background levels are significantly higher than those at GR0035A station. This suggests the influence of local emissions that are missing in the model or limitations due to the coarse model resolutions which can not represent this station.

At Bucharest, two high AOD peaks are modeled (Figure 3.5). Compared to AERONET ground observations, the first peak is well modeled on 23 July (0.46 against measurements 0.5) and underestimated on 30 July (0.51 against measurements 0.82 but simulated one day after the observations, probably due to uncertainty in the MODIS fire detection). The observed values of Ångström exponent are lower on 23 July ($\alpha \sim 0.57$), indicating a large fraction of coarse mode particles (probably dust transport), than on 30 July ($\alpha \sim 1.6$, large fraction of fine mode particles from fires). These under-estimations or over-estimations of the model's AOD, depending on the data set used for the evaluation (AERONET vs MODIS), underline uncertainties in AOD retrievals from measurements, which have already been observed by numerous studies (Li et al., 2009; Wu et al., 2017; Boiyo et al., 2017).

During the second fire event, the contribution from fires becomes predominant on 24 August (beginning of the event according to MODIS fire observations). The best agreement with $PM_{2.5}$ in Greece is obtained for the simulations *Poly-3km* and CHIMERE, although the latter overestimates surface concentrations on the following days. At the GR0035A station, the second observed peak during the second fire event is more than twice higher than the observed peak at the GR0039A and is measured one day earlier. In the simulations, the enhancement due to fires is similar in shape and magnitude, clearly highlighting the difficulty to simulate the exact temporal variability of emissions and transport of fire plumes in CTMs. Moreover, the station GR0035A is probably affected by dust episode during the second fire event.

For AOD at Lecce University, all simulations show good agreement to observations. According to AERONET level 1.5 measurements in Blida (in Algeria, sub-region MedReg5), an AOD peak (0.55) is observed on 27 August. The AOD simulated by CHIMERE and Polypohemus are consistent with the measurements mainly for *Poly-ref*, *Poly-3km*, *Poly-NoREM-PM$_{2.5}$* and *CHIMERE-ref*. *Poly-noI/S-VOCs* shows the lowest AOD values at Lecce University (0.38) and Blida (0.33). This suggests that taking into account I/S-VOCs emissions leads to higher and more realistic AOD at these stations.

In Blida, three peaks are simulated for 2 August, 6 August and 16 August. The observed values of the Ångström exponent are equal to $\alpha \sim 1.18$ and $\alpha \sim 1.14$ for the first and third peaks respectively, which indicates fine mode particles. Therefore, the first and third peaks are attributed mostly to fires in Algeria on 2 August and 16 August. However, emissions are probably underestimated, as all the models under-estimate the AOD fire-peaks. Since a lower value of the Ångström exponent is observed on 6 August ($\alpha \sim 0.91$), this second peak is probably attributed to dust.

This analysis highlights the strong regional impact during intense events on both AOD and $PM_{2.5}$ concentrations but also the difficulties in representing their amplitudes and variability. Considering the uncertainty on fire data and emissions ($\sim100\%$)

and on the observations (1-2% for AERONET observations (Eck et al., 1999) and ∼34% for AIRBASE observations (Bovchaliuk, 2013)), the performance obtained is considered very reasonable. The spread in the different model configurations tested shows the additional uncertainty on the modeling of fire impact. Observations generally lie within the simulated variability but

it is difficult to extract the best model configuration (it depends on the event and on the station).

For the first fire event, Figure 3.7 shows maps of the daily mean AOD at 550 nm from MODIS, modeled by Polyphemus (*Poly-ref*) and CHIMERE (*CHIMERE-ref*). The simulated AOD is generally in a good agreement with the observations in terms of localization. The AOD calculated from *Poly-ref* and *CHIMERE-ref* simulations is close to observations in Balkan. However, it seems overestimated in the fresh plume and further downwind (reaching ∼ 0.65 for *Poly-ref* and 0.52 for *CHIMERE-ref*). Results of *Poly-noI/S-VOCs* are close to those of the CHIMERE model, which does not include I/S-VOCs emissions. During the second fire event in Greece, the simulated AOD for *Poly-ref* and for *CHIMERE-ref* is about 1 and 0.9 respectively as shown in Figure D1 in Appendix D. The observed AOD can reach 0.9. However, both models overestimate AOD values in the fire

plume (reaching  0.98 for *Poly-ref* and 0.88 for *CHIMERE-ref* against 0.7-0.8 in Greece and 0.5 from MODIS). The fire plume is less pronounced in observations than in simulations.

Day-to-day comparisons for four selected days (24, 25, 27 and 29 August 2007) are shown in Figure E1 in Appendix E. The simulated AOD is consistent with the observations in terms of localization and general transport pathways. However, the simulated AOD is much higher in the Greek fires' plume compared with MODIS observations during the peak of emissions (25-29

August). This probably reflects too low temporal variability in the emissions. In the simulations, emissions are assumed constant during the day but comparisons suggest shorter temporal variability. This is also apparent in the time series of Figure 3.6, over MedReg2: the peak for the second fire event is twice longer in the simulations (double peak starting on the 25 August 2007) than in the observations. This peak over two days instead of one in the simulations suggests an overestimate of emissions during this event which is also observed with respect to surface observations in Greece. The first peak corresponds better to

observations. This shows that uncertainties are not only related to total emissions but also to their temporal variability and the associated transport pathways.

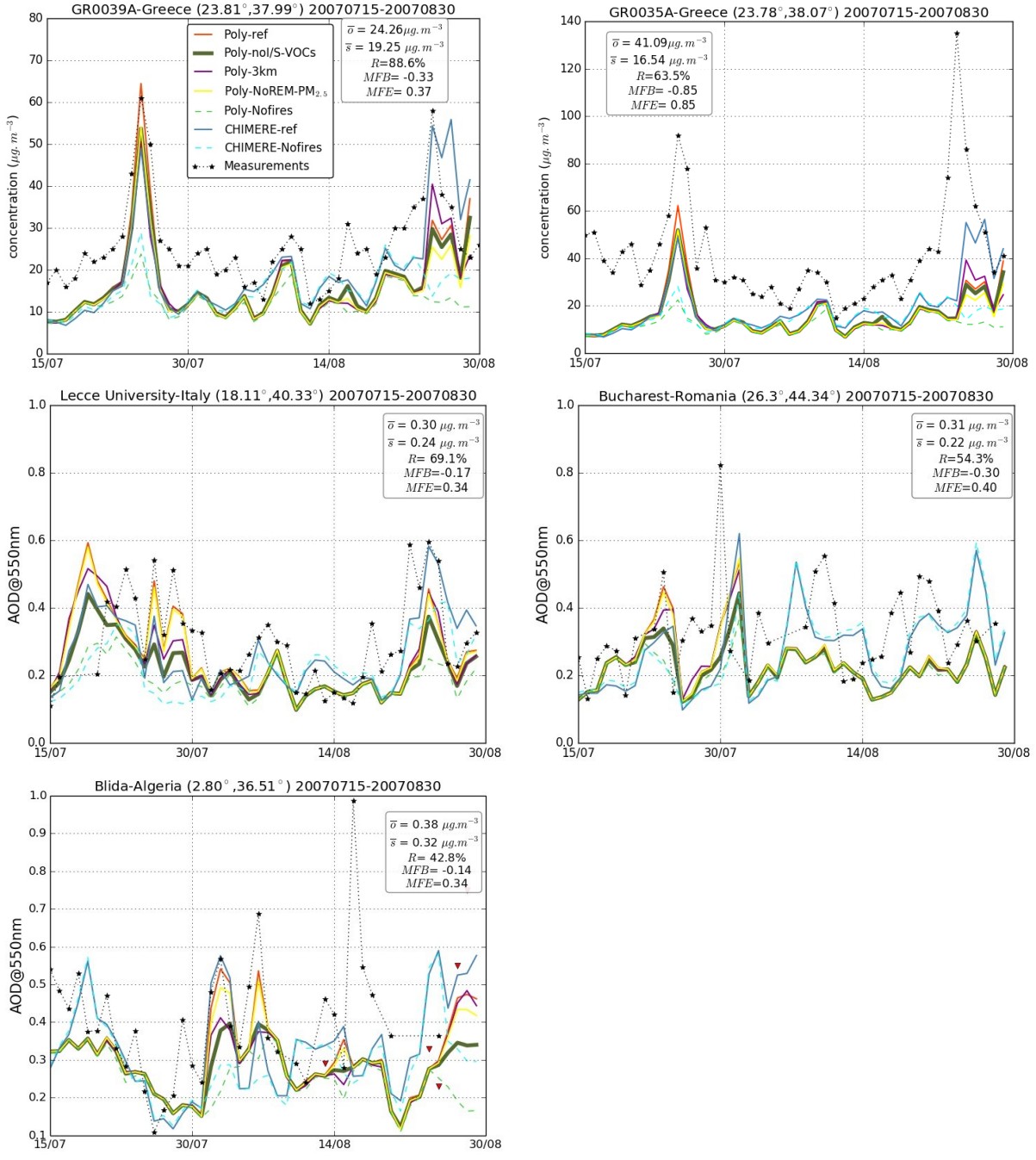

**Figure 3.5.** Time series from 15 July to 30 August of daily mean surface PM$_{2.5}$ concentrations at the AIRBASE stations GR0039A and GR0035A daily mean AOD at 550 nm at three AERONET stations (Lecce University, Bucharest, Blida). The red triangles in Blida station correspond to AERONET measurements using AOD level 1.5 AOD data at 500 nm. Statistics for simulation *Poly-ref* are shown at each station.

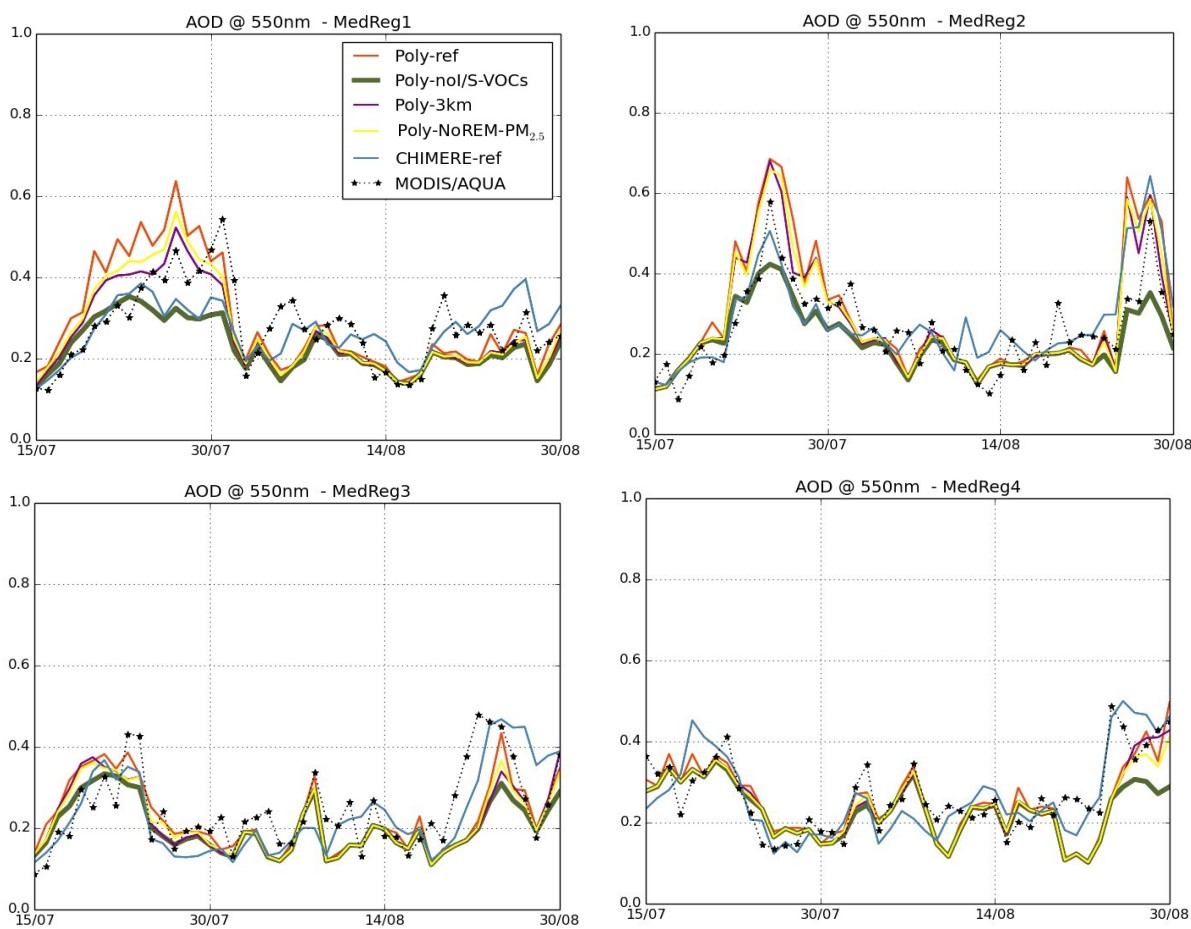

**Figure 3.6.** Daily mean AOD at 550 nm observed by MODIS and simulated by Polyphemus and CHIMERE from 15 July to 30 August 2007, in the sub-regions of Figure 2.1.

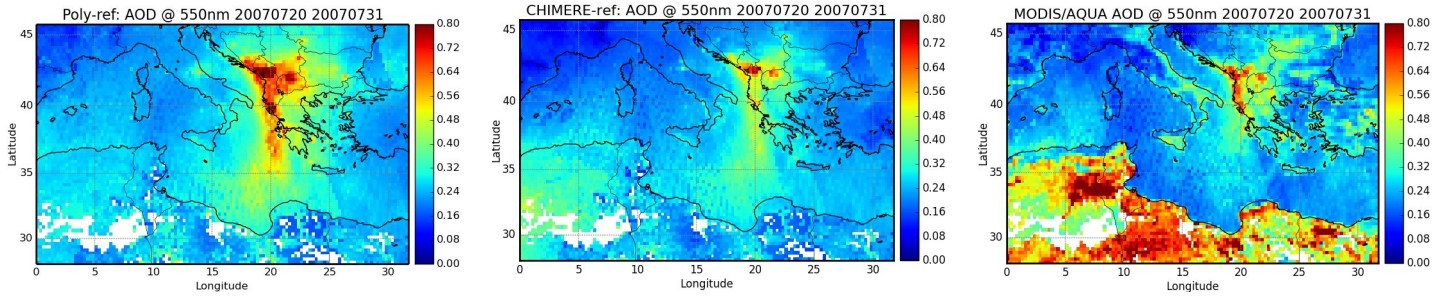

**Figure 3.7.** Mean total AOD (at 550 nm) from MODIS/AQUA, *Poly-ref* and *CHIMERE-ref* during the first fire event (20-31 July 2007).

## 4  Uncertainty and sensitivity analysis

The sensitivity of the modeling of I/S-VOCs emissions and injection heights on simulated surface PM$_{2.5}$ concentrations and
AOD is now evaluated regionally over the Mediterranean domain.

The sensitivity of the model results to the I/S-VOCs emissions and injection heights is compared to the inter-model sensitivity presented in Figure 4.1, which shows the relative differences between a sensitivity simulation (*CHIMERE-ref*, *Poly-noI/S-VOCs*, *Poly-3km*) and the reference simulation *Poly-ref*. To focus on fires impact, only the PM$_{2.5}$ concentrations and AOD exceeding 15 $\mu$g.m$^{-3}$ and 0.25 respectively, are taken into account when computing the relative differences between the
simulations. It is worthy to note that the arbitrary choice made in this work (injecting between 1 and 3 km) may overestimate the impact of injection height on surface PM$_{2.5}$ concentrations and underestimate it on long range transport (injecting fire emissions at or below 3 km remains conservatively low).

The inter-model sensitivity is low (relative differences below 20%) for both surface PM$_{2.5}$ concentrations and AOD, except in Balkan (sub-region MedReg1), where it can reach 50% locally. The high inter-model differences are slightly more spread
horizontally for surface PM$_{2.5}$ concentrations than for AOD. Furthermore, this region of high inter-model sensitivity corresponds to the region where the sensitivity to the injection height is the highest for PM$_{2.5}$ concentrations. It may therefore be linked to differences in the models' vertical discretisation. The models use different vertical coordinates and different numbers of vertical levels. The vertical resolution of the models is rather low as Polyphemus uses 14 vertical levels and CHIMERE uses 19 vertical levels.

Neglecting I/S-VOCs emissions leads to a decrease in surface PM$_{2.5}$ concentrations. The impact of I/S-VOCs emissions on surface PM$_{2.5}$ concentrations over the fire regions is mostly under 20%, but reaches 30% locally. The impact of I/S-VOCs emissions is spread over larger areas than the inter-model difference.

I/S-VOCs emissions have a higher impact on AOD than on surface PM$_{2.5}$ concentrations, since adding I/S-VOCs increases the total integrated PM$_{2.5}$ concentrations: the increase can be as high as 40% in Balkan, 30% in Greece and the fire plume.

As previously discussed, a surrogate species may be considered to represent a missing mass in the model fire emissions, REM-PM$_{2.5}$ in this study (emissions factors of PM$_{2.5}$ minus primary aerosols emissions of OM, BC and sulfate). This missing mass may be used to fill a gap in current evaluation of emissions. However, considering REM-PM$_{2.5}$ emissions and I/S/L-VOC

emissions may be redundant. Several models partly treat this missing mass by deducing I/S/L-VOCs contribution from particulate matter emissions (Koo et al., 2014; Konovalov et al., 2015). For example, several studies estimated the contribution from

I/S/L-VOCs emissions by multiplying the primary organic aerosols (POA) by a factor of 1.5 following chamber measurements (Robinson et al., 2007; Zhu et al., 2016; Kim et al., 2016). Some studies/models do not consider specific species/surrogates to treat these missing emissions, but simply use a ratio to reduce uncertainties related to the estimation of PM emissions. For example, Kaiser et al. (2012) use a factor of 3.4 for PM emissions based on the comparison between simulations and AOD observations.

Injecting above the boundary layer results in larger scale transport for $PM_{2.5}$ concentrations that leads to the highest impact

5   on surface $PM_{2.5}$ concentrations (40 to 50% near the fire regions and 30% in the fresh plume and further downwind). However, the impact of the injection height on AOD is lower, but still significant (mostly under 20%, but reaching 30% locally). Similar results are found in Turquety et al. (2007); Chen et al. (2009); Stein (2009); Daskalakis et al. (2015); Gonzi et al. (2015). In fact, previous studies highlighted the high sensivity of long range transport of carbon monoxide (CO) to wildfires injection height (Turquety et al., 2007; Gonzi et al., 2015). It also leads to a reduction of concentrations at the surface Chen et al. (2009).

10   The sensitivity analysis of Stein (2009) estimates a strong reduction in the surface $PM_{2.5}$ concentrations ($> 10 \, \mu g.m^{-3}$) caused by fire emissions injection height over the United States. According to Daskalakis et al. (2015), assumptions on the injection heights of fire emissions can also result regionally in up to 30% differences in the calculated tropospheric lifetime of pollutants. This can lead to significant interactions between isoprene and fire emissions. Daskalakis et al. (2015) showed that these interactions affect the effectiveness of isoprene to produce secondary aerosols (up to 18%).

This analysis highlights that injecting above the boundary layer is more critical for surface $PM_{2.5}$ concentrations than integrating I/S-VOCs emissions, since 78% of fire emissions are emitted above the boundary layer. However, for AOD and

vertically integrated concentrations, integrating I/S-VOCs emissions is more critical than the injection heights.

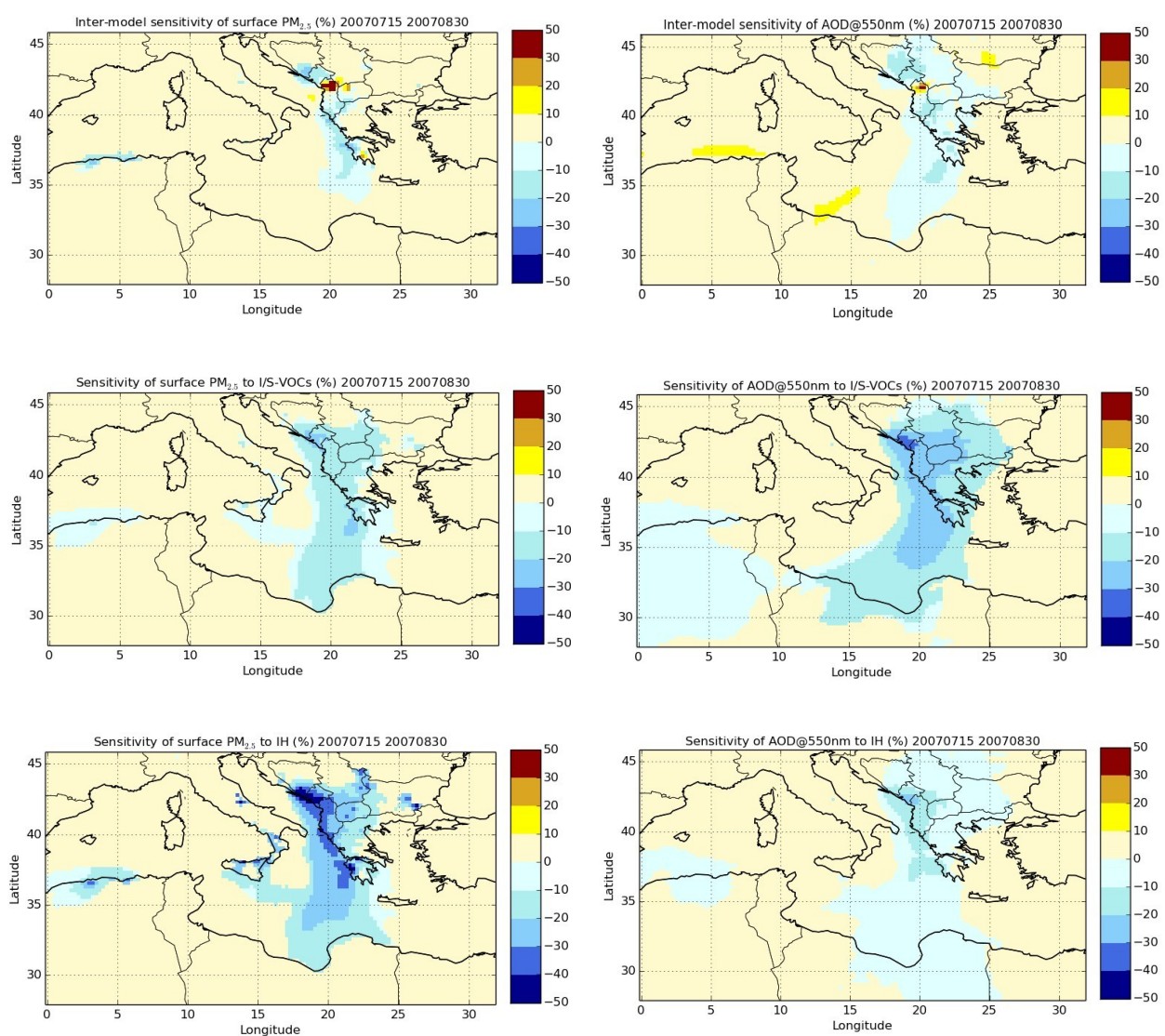

**Figure 4.1.** Sensitivity of surface PM$_{2.5}$ concentrations (left panels) and AOD at 550 nm (right panels) to the CTM used (*CHIMERE-ref* or *Poly-ref*, upper panels), the modelling of I/S-VOCs emissions (*Poly-noI/S-VOCs* or *Poly-ref*, middle panels) and the injection height (IH) (*Poly-3km* or *Poly-ref*, lower panels) during the summer 2007 (15 July to 30 August). The simulation *Poly-ref* is used in all panels to estimate the relative differences.

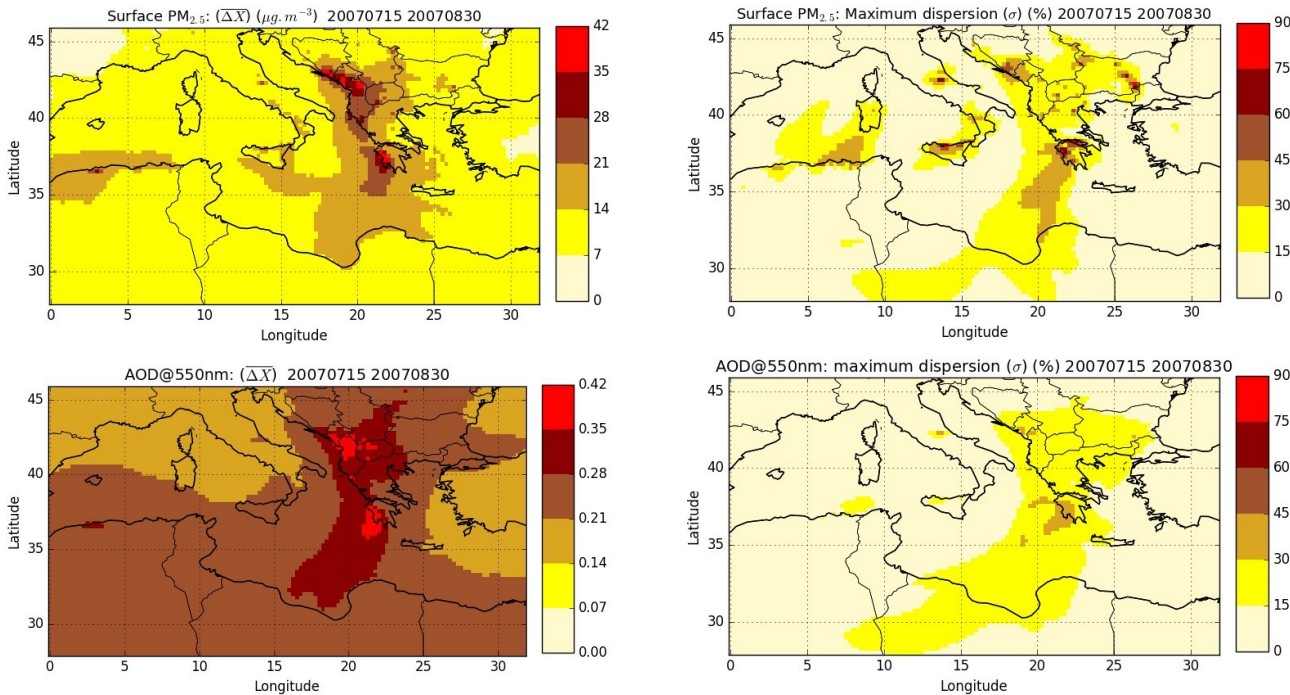

**Figure 4.2.** Mean surface PM$_{2.5}$ concentrations and AOD (estimated using four Polyphemus sensitivity simulations) (left panel) and their relative maximum statistical dispersion ($\sigma$) (right panel) during the summer 2007.

The ensemble of the sensitivity simulations provides an estimate of the uncertainty of the modeling of fire plumes. To quantify uncertainties related to fire emissions modeling, the maximum statistical dispersion ($\sigma$) is used as a statistical estimator. The maximum statistical dispersion ($\sigma$) for surface PM$_{2.5}$ and AOD calculated as:

$$\sigma = \frac{\sqrt{\frac{1}{N}\sum_{i=1}^{N}(X_i - \overline{\Delta X})^2}}{\overline{\Delta X}}.100 \tag{5}$$

$$\overline{\Delta X} = \frac{1}{N}\sum_{i=1}^{N}(X_i) \tag{6}$$

with $X$ refers to either surface PM$_{2.5}$ concentrations or AOD, $N$ is the total number of the simulations with fire emissions included ($N$ is equal to 4).

Figure 4.2 shows the average surface PM$_{2.5}$ concentrations and AOD estimated using four Polyphemus simulations and the maximum statistical dispersion ($\sigma$) for surface PM$_{2.5}$ concentrations and AOD. The maximum statistical dispersion related to simulated surface PM$_{2.5}$ concentrations and AOD is higher near the fire regions. The mean surface PM$_{2.5}$ concentrations

estimated from the four Polyphemus simulations can reach 42 $\mu$g.m$^{-3}$ in Balkan and Greece with a statistical dispersion that can be as high as 75%. Lower mean surface PM$_{2.5}$ concentrations are calculated for Algeria, in the fresh fire plume and further downwind (28 $\mu$g.m$^{-3}$ with a maximum statistical dispersion reaching 45%). The mean AOD estimated as the average of four Polyphemus simulations can reach 0.42 in Balkan and Greece, 0.36 in the fire plume with a statistical dispersion that can reach 45% and 36% respectively.

Although these uncertainties are quite high, they are probably still underestimated since several other sources of uncertainties should be considered. First of all, uncertainty on the initial emissions is important. Uncertainty on the burned area and the associated temporal evolution (used as input to the calculation of emissions) is also high. Giglio et al. (2010) found that uncertainties on MODIS observation can reach about 5 days mainly due to cloud cover.

Although the contribution from organic aerosols is dominant, biomass burning is also a source of inorganics. Over Europe,
inorganics (mainly sulfate, sea salt and ammonium) contribute highly to PM$_{2.5}$ composition, when fire emissions are not considered (e.g., Fountoukis et al. 2011; Chrit et al. 2018). However, during fire events the contribution of inorganics is lower than the contribution of organics (8 to 9% from inorganics against 40 to 80% from organics). Focusing on this inorganic contribution from fires, sulfate, ammonium and nitrate are the predominant inorganic components. The formation of inorganics due to wildfires is found to be low compared to the formation of organics. However, our simulation takes into account emissions of inorganic precursors such as ammoniac (NH$_3$) with emission factors from Akagi et al. (2011). Several studies (R'honi et al., 2013; Van Damme et al., 2014; Whitburn et al., 2017), show that large emissions of NH$_3$ are released by biomass burning. Whitburn et al. (2017) studied the enhancement ratios NH$_3$/CO for biomass burning emissions in the tropics using observations from the IASI satellite based instrument. They found a significant variability due to fire contribution. According to Whitburn
et al. (2017), the emission ratios NH$_3$/CO in the tropics derived from IASI observations (as in Van Damme et al. (2014)) are rather on the lower end of those reported in Akagi et al. (2011) that are used here. If fire emissions are important for the regional budget of organics, more observations are required to provide emission factors of NH$_3$ and concentrations of inorganics should be evaluated close to fire regions.

Deposition can be considered as a source of uncertainty on PM and AOD over the Euro-Mediterranean region during summer
2007. Roustan et al. (2010) pointed out the importance of dry and wet deposition over Europe while studying the sensitivity of Polyphemus to input data over Europe with a focus on aerosols. They found that PM is sensitive to options influencing deposition such as wet diameter and aerosol density. They found that during both summer and winter, the uncertainties on wet diameter and aerosol density can reach 19% and 9% respectively. Several studies also highlight the importance of the gas-phase deposition of I/S/L-VOCs (Hodzic et al., 2016; Knote et al., 2015; Karl et al., 2010; Bessagnet et al., 2010; Hallquist et al.,
2009). Hallquist et al. (2009) highlighted the importance of the vapor deposition which is higher than the particle deposition (800 vs 150 TgC/yr). Knote et al. (2015) showed that the gas-phase I/S/L-VOCs that are highly water soluble (Henry's law constants H$^*$ between 10$^5$ and 10$^{10}$ M.atm$^{-1}$) are very susceptible to removal processes in the atmosphere (wet deposition and dry deposition). Ignoring the removal of gas-phase I/S/L-VOCs (dry/wet deposition) in the models can lead to uncertainties on SOA concentrations, AOD and PM$_{2.5}$ concentrations.

Meteorological conditions also play an import role in pollution dispersion and the capabilities of models to reproduce observed pollution plumes. Garcia-Menendez et al. (2014) found that simulated $PM_{2.5}$ concentrations at urban sites displayed large sensitivities to wind perturbations within the error range of meteorological inputs. Rea et al. (2016) added that special attention must be paid to the PBL height, which can have a considerable impact on the fire emissions injection heights. Therefore an assessment of uncertainties related to meteorological data should be investigated.

## 5    Air quality exceedances

Air quality impact is evaluated by analyzing the number of threshold exceedances during the summer of 2007. It corresponds to the number of times daily averaged $PM_{2.5}$ surface concentrations exceeds 25 $\mu$g.m$^{-3}$ (World Health Organisation recommendation, (Krzyzanowski and Cohen, 2008)).

The number of exceedances predicted by the model is first compared to exceedances observed by AIRBASE stations. Table F1 in Appendix F presents the modeled and observed $PM_{2.5}$ exceedances at each AIRBASE stations during the whole summer 2007 and during fire events. Generally, the models (Polyphemus and CHIMERE) underestimates $PM_{2.5}$ air quality exceedances because the horizontal resolution used here does not allow the representation of local pollution (especially for the station GR0035A). Better performance is observed during fire events than the whole period. Near fire regions, at the stations GR0039A, the number of exceedance is well-modeled, in spite of the slight underestimation during both fire events compared 15    to the observed $PM_{2.5}$ exceedances. However, at the GR0035A station, the underestimation of $PM_{2.5}$ exceedances is sharp. This can be explained by the strong underestimation of the background $PM_{2.5}$ levels as shown in Figure 3.5. Far from fire regions, the $PM_{2.5}$ exceedances modeled by Polyphemus are in a good agreement with the observed ones especially during the two fire events. The simulated $PM_{2.5}$ exceedances by CHIMERE in the fire plume are overestimated at some stations.

Figure 5.1 shows the additional days with $PM_{2.5}$ exceedances due to fires simulated by Polyphemus (difference between the 20    *Poly-ref* and *Poly-Nofires*). Most are concentrated around fire sources, mainly in Balkan (30 days). Figure G1 in Appendix G shows that conclusions for CHIMERE are similar to those for Polyphemus. Fires cause up to 49.5% of the total simulated exceedances from 15 July to 30 August 2007.

During the fire periods, surface $PM_{2.5}$ concentrations simulated in *Poly-ref* in MedReg1 and MedReg2 sub-regions are composed mainly of OM (54% with 61.6% due to fire for MedReg1 and 52.3% with 60.1% due to fires for MedReg2), and 25    inorganics (27% with 8% due to fires for MedReg1 and 15% with 9% due to fires for MedReg2).

Figure 5.1 shows that the maximum statistical dispersion on the simulation of $PM_{2.5}$ exceedances can reach 15 days in regions affected by fires, particularly 5 days in GR0039A, 12 days in the fire plume and 6 days further downwind, based on our ensemble of sensitivity simulations.

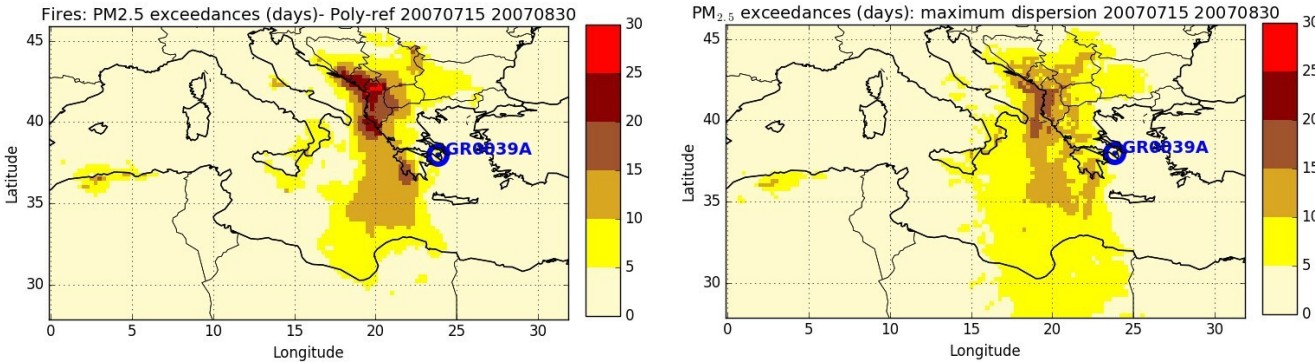

**Figure 5.1.** The additional days with PM$_{2.5}$ exceedances due to fires simulated by Polyphemus (difference between the *Poly-ref* and *Poly-Nofires*) (left panel) and the maximum statistical dispersion ($\sigma$) related to fire emissions for additional days of PM$_{2.5}$ exceedances due to fires (right panel) during the summer 2007 (from 15 July to 30 August 2007).

## 6  Conclusions

A sensitivity analysis was undertaken to estimate the uncertainty of the parameterization of wildfire emissions into air quality models over the Euro-Mediterranean region during the summer 2007. The ability of Polyphemus/Polair3D and CHIMERE to simulate regional surface PM$_{2.5}$ concentrations and aerosol optical depth (AOD) was evaluated based on comparisons to available measurements. The comparison of surface PM$_{2.5}$ concentrations and AOD at 550 nm to available surface measurements (background suburban and rural stations only) shows that both models meet the performance criteria. A clear improvement is noticed when biomass burning emissions are taken into account.

The contribution of biomass burning is large in Balkan and Easten Europe during the first fire period (20–31 July) and Greece and Algeria during the second period (24–30 August). According to the simulations, PM$_{2.5}$ close to regions affected by fires is mostly composed of organic aerosol (47% to 85%), with a strong contribution from I/S/L-VOCs (46% to 80%). However, only two AIRBASE stations (GR0039A and GR0035A in Greece) and three AERONET stations (Lecce University in Italy, Blida in Algeria and Bucharest in Romania) show fire contribution higher than 10% according to the model simulations. The lack of surface observations strongly limits this evaluation but it is partly complemented by comparisons to MODIS satellite-based observations of AOD. Comparisons to satellite observations over subregions show a good simulation of the daily variability of AOD, with high correlation coefficients for Polyphemus ($\sim$90% on average) and CHIMERE (80% on average). Comparisons to surface and remote sensing observations show that the models can simulate enhancements of a good order of magnitude and +/- 1 day uncertainty in the timing.

Two critical parameters, SOA formation from I/S-VOCs and injection heights, are considered as the two main sources of uncertainties in the calculation of wildfires impact on surface PM$_{2.5}$ concentrations and AOD. Sensitivities to these key parameters are computed using simulations performed with different configurations of Polyphemus. These configurations are chosen to maximize the sensitivities.

Compared to satellite observations, the AOD modeled in simulations including I/S-VOCs is overestimated during the fire events, by about 30% for *Poly-ref* and *Poly-3km* and 10% for *Poly-NoREM-PM$_{2.5}$*, mainly in the two sub-regions MedReg1 and MedReg2, closest to fire emissions. Since unidentified primary particles (REM-PM$_{2.5}$) emitted from biomass burning are not considered in *Poly-NoREM-PM$_{2.5}$*, the simulated AOD values are closer to MODIS observations. This suggests that REM-PM$_{2.5}$ could correspond to I/S-VOCs.

Comparisons to AIRBASE measurements show a good simulation of the surface PM$_{2.5}$ concentrations during the first fire event (at GR0039A in Greece, sub-region MedReg2). The reference simulation (*Poly-ref*) shows closest comparisons (with a high correlation coefficient, >88%, and a low bias), while PM$_{2.5}$ peaks are slightly underestimated by all the other simulations. However, the measured AOD tends to be underestimated by all model simulations at Bucharest (Romania, sub-region MedReg1). During the second fire event, surface PM$_{2.5}$ concentrations in simulations *Poly-3km* and *CHIMERE-ref* are in good agreement with AIRBASE measurements, but are underestimated in other simulations. The modeled AOD is well simulated compared to AERONET observations at Lecce University and Blida by *CHIMERE-ref* and all the Polyphemus simulations except for *Poly-noI/S-VOCs*, which shows low AOD values at these stations. This suggests that taking into account I/S-VOCs improves the simulated AOD values at these stations. In spite of the uncertainty on fire emissions (>100%) and on observations, this analysis shows that the models succeed to reproduce the PM$_{2.5}$ concentrations and AOD during such large wildfire event.

Further analysis of uncertainties is conducted at regional scale based solely on the set of sensitivity simulations conducted with the Polyphemus model. AOD is particularly sensitive to I/S-VOCs emissions (up to 40% sensitivity), while surface PM$_{2.5}$ concentrations are particularly sensitive to the injection heights (up to 50% sensitivity). These sensitivities are most of the time higher than inter-model sensitivities, which are mostly linked to the model vertical discretization close to fire emissions. The statistical dispersion of the ensemble of simulations based on different configurations of Polyphemus is used to evaluate the maximum uncertainty on surface PM$_{2.5}$ concentrations and AOD associated with these two parameters. During the summer 2007, the maximum statistical dispersion ($\sigma$) is as high as 75% for surface PM$_{2.5}$ in the Balkans and Greece and varies between 36 and 45% for AOD above fire regions. The number of daily exceedance of WHO recommendation of 25 $\mu$g.m$^{-3}$ for PM$_{2.5}$ 24-hour mean reaches 30 days for the fire region and 10 – 15 days for the fire plume over the simulated period of 46 days. The maximum statistical dispersion on this indicator is large, reaching 15 days close to fires and 5-10 in the fire plume.

Although relatively high, this estimate of uncertainty is very conservative since many other parameters may alter the quality of the simulations of wildfires impact on atmospheric composition. In addition to uncertainty on emissions (initial fire characteristics, vegetation type and fraction burned, emission factors, etc.) and SOA formation, the formation of inorganic aerosols and the uncertainty on meteorological conditions (pyroconvection but also general stability in the region) and on deposition are a few examples of processes that may alter the simulated aerosol budgets. More integrated surface and in situ observations would be necessary for a precise evaluation of emissions, the simulated long-range transport from fire emissions, the aerosol speciation within the plumes and the resulting impact on air quality.

*Acknowledgements.* CEREA and LMD are members of the Institut Pierre-Simon Laplace (IPSL). This research was partially funded by a PhD grant from École des Ponts ParisTech, and by the french ministry of ecology, by CNRS-INSU (GMES-MDD program, NATORGA project). This study also received financial support from MISTRALS by ADEME, CEA, INSU, and Meteo-France, as part of the ChArMEx project. We are grateful to the EEA for maintaining and providing the AirBase database (more particularly for the GR0039A station). We thank the principal investigators and their staff for establishing and maintaining the AERONET sites used in this study: Brent Holben for Blida, Maria Rita Perrone for Lecce University and Doina Nicolae for Bucharest Inoe. We wish to thank the National Aeronautics and Space Agency (NASA) for the availability of the MODIS data. Finally, we thank the anonymous reviewers and editors for their comments which have helped improve the manuscript.

## Appendix A

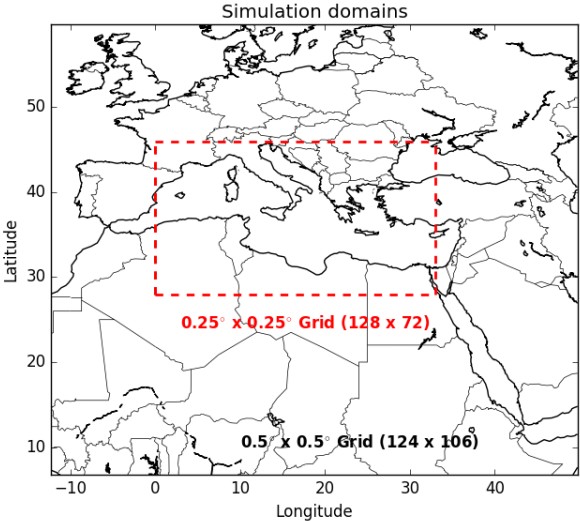

**Figure A1.** Simulation domains including one large domain (with a 0.5°×0.5° horizontal resolution) and a smaller domain (at a 0.25°×0.25° horizontal resolution) delimited by the dotted red box.

## Appendix B

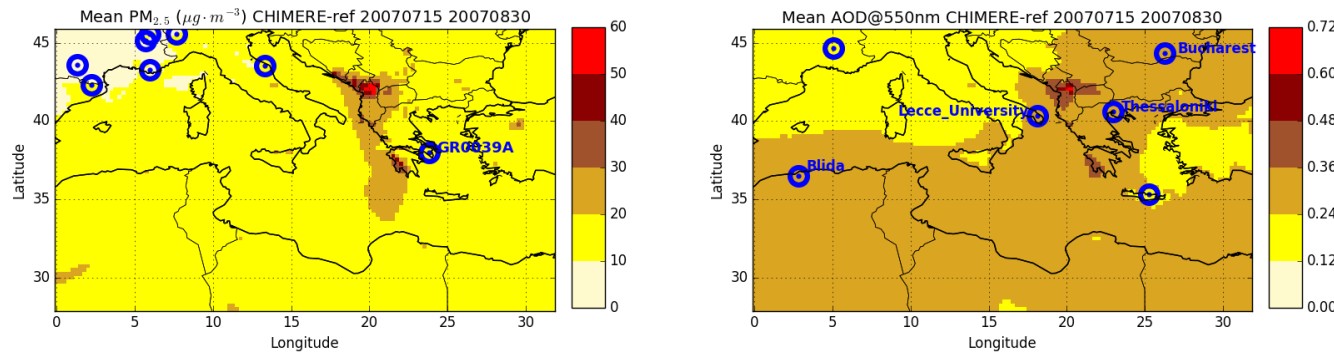

**Figure B1.** Daily mean surface PM$_{2.5}$ and AOD at 550 nm from the *CHIMERE-ref* simulation averaged over the summer of 2007 (the 8 AIRBASE and 6 AERONET stations, used in this work, are represented here in blue dots).

# Appendix C

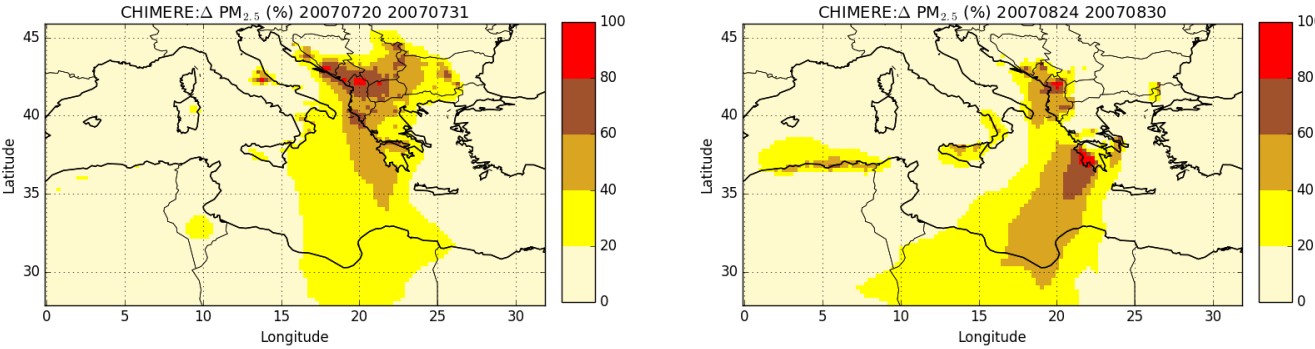

**Figure C1.** Left panel: relative difference of surface PM$_{2.5}$ concentrations between simulations *CHIMERE-ref* and *CHIMERE-Nofires* during the first fire event. Right panel: relative difference of surface PM$_{2.5}$ concentrations between simulations *CHIMERE-ref* and *CHIMERE-Nofires* during the second fire event.

# Appendix D

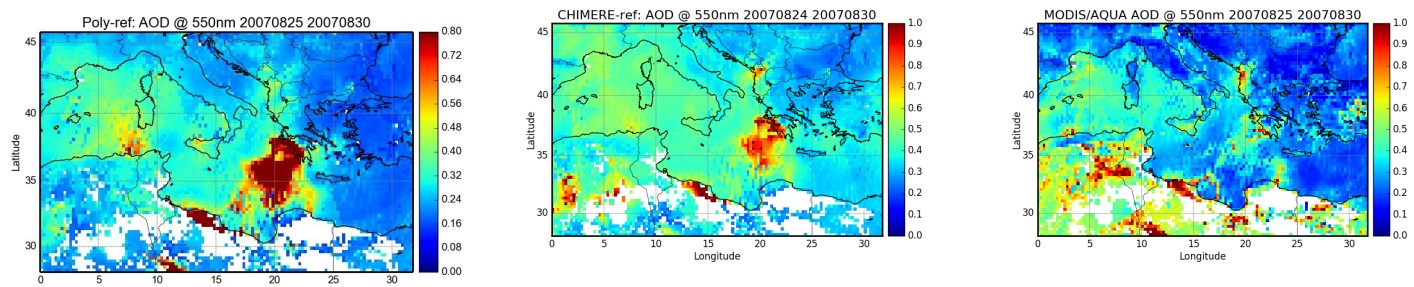

**Figure D1.** Mean total AOD (at 550 nm) from MODIS/AQUA, *Poly-ref* and *CHIMERE-ref* during the second fire event (24-30 August 2007).

**Appendix E**

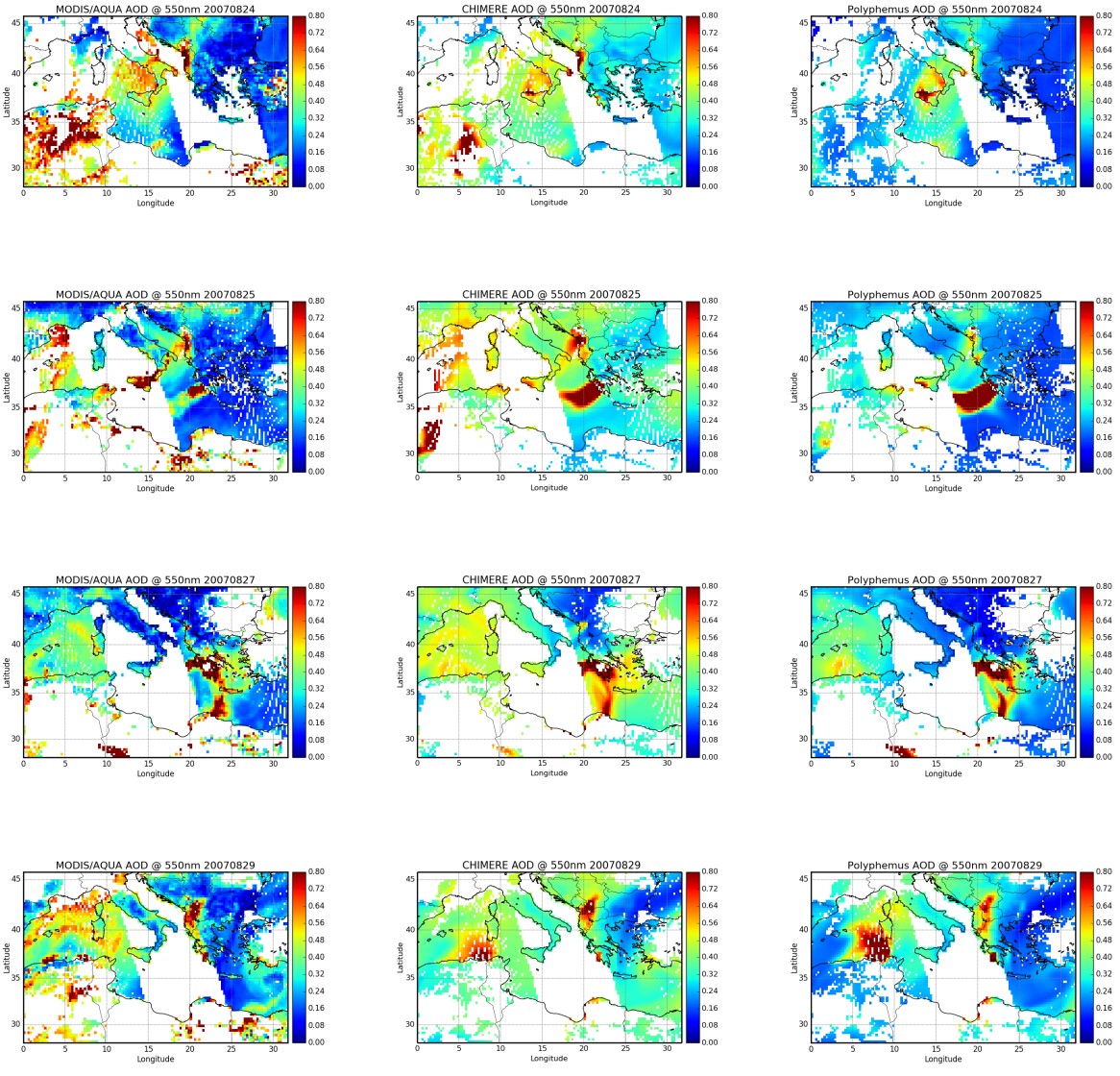

**Figure E1.** Total AOD (at 550nm) from MODIS/AQUA and the corresponding Polyphemus AOD (*Poly-ref*) and CHIMERE AOD (*CHIMERE-ref*) for four selected days (24, 25, 27 and 29 August 2007).

# Appendix F

**Table F1.** Modeled and observed PM$_{2.5}$ exceedances at each AIRBASE stations. Values between brackets correspond to modeled PM$_{2.5}$ exceedances for simulations without fire emissions (*Poly-Nofires* and *CHIMERE-Nofires)*.

| AIRBASE stations | Observed PM$_{2.5}$ exceedances | | | Modeled PM$_{2.5}$ exceedances *Poly-ref* | | | Modeled PM$_{2.5}$ exceedances *CHIMERE-ref* | | |
|---|---|---|---|---|---|---|---|---|---|
| | Whole period | First event | Second event | Whole period | First event | Second event | Whole period | First event | Second event |
| GR0039A | 14 | 5 | 5 | 7(0) | 3(0) | 4(0) | 9(2) | 3(1) | 5(0) |
| GR0035A | 37 | 12 | 7 | 7(0) | 3(0) | 4(0) | 9(2) | 3(1) | 5(0) |
| IT0459A | 4 | 2 | 2 | 0(0) | 0(0) | 0(0) | 0(2) | 0(0) | 0(0) |
| FR12021 | 0 | 0 | 0 | 0(0) | 0(0) | 0(0) | 2(2) | 0(0) | 2(2) |
| ES0010R | 0 | 0 | 0 | 1(1) | 0(0) | 1(1) | 2(2) | 0(0) | 2(2) |
| FR15043 | 0 | 0 | 0 | 0(0) | 0(0) | 0(0) | 4(4) | 0(0) | 4(4) |
| FR33101 | 0 | 0 | 0 | 0(0) | 0(0) | 0(0) | 3(3) | 0(0) | 3(3) |
| FR03043 | 5 | 0 | 3 | 2(2) | 0(0) | 2(2) | 4(4) | 0(0) | 4(4) |
| FR33102 | 0 | 0 | 0 | 0(0) | 0(0) | 0(0) | 3(3) | 0(0) | 3(3) |

**Appendix G**

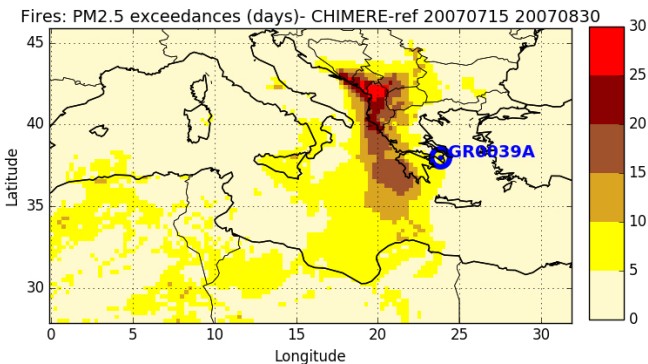

**Figure G1.** The additional days with PM$_{2.5}$ exceedances due to fires simulated by CHIMERE (difference between the *CHIMERE-ref* and *CHIMERE-Nofires*) during the summer 2007 (from 15 July to 30 August 2007).

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
