# Peer review of "Impact of wildfires on particulate matter in the Euro-Mediterranean in 2007: sensitivity to some parameterizations of emissions in air quality models"

_Atmospheric Chemistry and Physics, 2018_

## Referee Comment (RC1) · Anonymous Referee #1 · 19 Jun 2018

General comments

The manuscript addresses an interesting subject relevant with the impact of wildfire emissions on air quality with a focus on the atmospheric PM levels. The underestimation of PM concentrations by air quality models is a well-known issue becoming even more evident in extreme events like those of biomass burning and wildfires, considering that the estimation of amounts of pollutants emitted during these events has many uncertainties while the physical phenomena to describe their evolution (e.g. pyroconvection to determine injection height) are poorly described in the air quality models.

[Figure]

The submitted paper can only be considered for publication after major revisions. The main general comment for the manuscript is that several of the results presented can be considered as tentative since they are provided for and refer to a single site (GR0039A). In addition, in the manuscript more results are presented for the first summer period wildfire event (20-31 July 2007) although the maximum of emissions is during the second summer period wildfire event (24-30 August 2007) as shown in Figure 2.

Specific comments

1. Page 6, lines 20-22: Instead of primary aerosol oxidation, are you probably referring to the oxidation of I/S/L-VOCs by OH in the gas phase?

2. Page 8, lines 8-9: Please use a reference and explain why POA emissions taken into account are modelled as LVOCs.

3. Pages 11-12, Section 3.3: The overall performance of CHIMERE model is shown to be slightly better than Polyphemus model considering also the statistics in Tables 3 and 4. Under this view, you have to present results of CHIMERE model in Figures 3, 4 and 11 additional to those of Polyphemus model for comparison.

4. Pages 12, Figure 3: Presenting similar maps using the data of MODIS is necessary to support better the comments in the manuscript about Figure 3.

5. Page 13, Section 3.4.1: Figure 4 reveals several parts of the Mediterranean area being affected by the wildfire plumes for example Central and South Italy, Sicily, Bulgaria, almost the whole country of Greece etc. In the manuscript you are referring to only 2 stations being affected by the wildfire emissions during the first event and 3 stations during the second one. Aren't there additional AIRBASE or AERONET stations that could be used to support your analysis? (for example GR0035A is also a background suburban station where PM2.5 concentrations are being measured; is there a reason for not using measured data from this station?). Is the performance of the model good also for the remaining areas that are not affected by the fires?
6. Page 14-15, Section 3.4.2: The results presented in this section refer only to a single station and only to the fire event of the first period. Under this view, they are of very limited reference and cannot be used to characterise the aerosol composition in the Mediterranean areas affected by the fire plumes. Similar results with those presented in Figure 5 have to be presented for different stations and wider areas affected by the fire emissions and separately for both fire peak events in order possible differences to be identified.

7. Page 16, lines 4-8: Comments for Figure 8 is better to be written at the end of the section 3.4.3 after the comments for Figures 6 and 7.

8. Page 16, Figure 8: Please present and comment on similar maps based on the models and MODIS results for the second fire event in August.

9. Page 23, Section 4.1: In lines 10-12, it is commented that according to the Polyphemus model results the number of days, when the WHO air quality limit for the daily PM2.5 values is exceeded in GR0039A station, is 7 while the corresponding number based on measurements is 14. This corresponds to a -50% bias by the model. Is it acceptable? No data are provided for additional stations in the study area rendering the results presenting in this section very tentative. Further evidence is necessary to prove the acceptable performance of Poly-ref runs (and CHIMERE-ref runs, please see a previous comment) in the estimation of the PM2.5 air quality exceedences supporting better the necessity for the maps in the Figure 11 to be presented.

10. Page 23, Figure 11: Figure 11 is not clearly explained in the manuscript. What is presented in Figure 11 (left panel)? Is it the additional days with exceedences due to fires (i.e. (days with exceedences Poly-ref) minus (days with exceedences Poly-Nofires))?

11. Page 23, lines 16-18: It is commented by the authors that PM2.5 concentrations are composed mainly of organic matter. However adding the percentages in the parentheses (i.e. 7.3% for POA and 19.1% for SOA), a percentage total contribution of 26.1%

is estimated. Is this contribution to PM2.5 the main? Please clarify.

12. Page 24, lines 9-10: The percentages have been estimated for one site and for one fire event and have a very limited reference (spatial and temporal). Please see a previous comment of mine so as to update the conclusions.

13. Page 24, lines 10-12: Which Figure or Table supports these high correlation values and where in the manuscript are these results commented? Is it Figure 7?

14. Page 24, lines 14-16: This conclusion is not straightforward and requires further clarifications not only in conclusions but also in the manuscript. In addition, it is not clear how overestimations in simulated results may indicate missing emissions.
* * *

---

## Referee Comment (RC2) · Anonymous Referee #2 · 28 Jun 2018

This paper tries to evaluate model estimates of fire-related PM concetrations through evaluation with data from surface networks and satellite remote sensing. However, I think this work is flawed due to the decision to use only PM measurements. For some reason they do not make use of the rather extensive measurements which are available (Airbase,EMEP, or even AMS data) of sulfate, nitrate, ammonium, or other inorganic components.

This is a serious problem I think. For example, Fountoukis et al (2011) suggested significant contributions of sulfate, nitrate and ammonium to PM1 mass, and backed

up their analysis with comparison to measurements. Van Damme et al. (2014) showed enormous levels of NH3 from biomass burning, which presumably caused high PM2.5 also.

Further, on p10 they claim that by focusing on PM2.5 they avoid contributions from dust, but this simply isn't the case. The low-diameter tail of coarse-mode dust (and sea-salt) contributes of course to PM2.5, and measurements in southern Europe have shown that this can be a significant fraction even over long-periods (Putaud et al., 2010). (This same publication also demonstrates clearly the importance of non-organic compounds in contributing to PM2.5.)

I see no reason to believe that changes in model performance attributed here to various assumptions on OA emissions could not be masked or mistaken for problems with inorganic pollutants. As well as attending to the points raised below, I would encourage the authors to do a thorough analysis of the model performance for all the major compounds, and across many sites. I am afraid I cannot recommend this paper for publication without major revision.

Other points

0. The title is actually quite misleading. The paper only addresses some aspects of OM emissions, not emissions in general.

1. The word dispersion, used in eg line 17 of the abstract, is usually associated with turbulent mixing in the air pollution context. Please use another term to make clear that here you mean statistical spread.

2. p2, L20. Give reference for statements about anthropogenic origin.

3. p3, L8. Give reference for the volatility limits and definitions. Also, do you really mean from 1e4, or from 0.32e4 (see eg Table 4 of Murphy et al., 2014) ?

4. Units (e.g. p3. L8) should not be in italics.

5. p3, L23. Change 'current CTMs' to 'two CTMs'. Your study says little about the many other CTMs used in Europe.

6. Table 1 is very vague and uses many acronyms that haven't been explained yet. What is ECMWF, which version? What does 'from nesting simulations' mean? Which version of MEGAN? Which version of EMEP emissions? Give more specific details and references here.

7. P6, L3. The nesting factor is here given as 2:1, but usually one uses an odd ratio to avoid interpolation errors (e.g. https://earthscience.stackexchange.com/questions/8067/why-is-wrf-most-often-configured-at-31-nesting-ratio). Have the authors checked if their results are robust?

8. p6, L9. Biogenic emissions of what? I assume isoprene and terpenes, but which?

9. p6, L17-24. The explanation of the VBS scheme used is unclear. Firstly, where do the numbers -0.04, 1.93 and 3.5 come from? They are not suggested by May et al. 2013, and not obvious to the reader. Secondly, L21 suggests that OH+POA leads to SOA compounds in the same VBS bin, but L22-23 suggests a reduction in volatility equivalent by a factor of 100. Which is it?

10. p6, L17. The use of 'log' is ambiguous. In most VBS papers it means log10, but in other papers (especially mathematical) and common programming languages (eg python) it means the natural logarithm, $\log_e$ (or ln). I suggest using log10 to be clear.

11. p6, L33-34. Akagi provides emission factors for several ecosystems, but non are a close match to Mediterranean landscapes. And some of these emitted isoprene and terpenes, which also form SOA. Which emissions were in fact used?

12. Also, L33 suggests that only carbon emissions were used, but many studies suggest that fires are a major and sometimes huge source of gases including $NH_3$ (eg van der Werf et al., 2017, Van Damme et al., 2014), and hence of inorganic PM. Were such

emissions examined for their impact on PM? Were they checked?

13. p7, on. The notation PPMfine is very confusing. This is usually used to represent emissions of any primary particulate matter including OM, BC and inorganics. Am I right in saying that your PPMfine is PM2.5 - OC - BC - SIA? In which case I would expect most of be essentially OM-OC - is that what you mean? To further the confusion with this PPMfine notation, you introduce PPMcoarse which is more traditional, just PM10-PM2.5. So, PPMfine excludes OC, BC and SIA, and consists largely of O and H, whereas PPMcoarse consists of OM+BC+SIA. As I said, confusing.

14. p7, Figure 1. It is difficult to see the yellow fire labels on top of the yellow background. Change the colors.

15. p9. Table 2.

- The Poly-ref emissions are said to be introduced between 1km and the PBL. I assume you mean between ground level and min(PBL,1km)? If not, I have a serious problem with your base-case!

- What is PB?

References: —————

Fountoukis, C. et al., Evaluation of a three-dimensional chemical transport model (PM-CAMx) in the European domain during the EUCAARI May 2008 campaign Atmos. Chem. Physics, 2011, 11, 10331-10347

Murphy, B. N. et al., A naming convention for atmospheric organic aerosol Atmos. Chem. Physics, 2014, 14, 5825-5839

Putaud, J. P. et al., A European aerosol phenomenology-3: Physical and chemical characteristics of particulate matter from 60 rural, urban, and kerbside sites across Europe Atmos. Environ., 2010, 44, 1308-1320

Van Damme, M. et al., Evaluating 4 years of atmospheric ammonia (NH3) over Europe

using IASI satellite observations and LOTOS‐EUROS model results, J. Geophys. Res. Atmos., 119, 2014, 9549–9566, doi: 10.1002/2014JD021911.

van der Werf, G.R. et al., Global fire emissions estimates during 1997-2016 Earth System Sci. Data., 2017, 9, 697-720

---

## Referee Comment (RC3) · Anonymous Referee #3 · 4 Jul 2018

General comments

This paper is about the modeling of wildfires in air quality models and the associated uncertainties on the aerosols concentrations and optical properties. The article is based on several numerical simulations of the 2007 fires in the Mediterranean region with the Polyphemus and CHIMERE models. Simulations explore the sensitivity of these models to chemical related factors (mainly VOC emissions) and dynamics (injection height).

[Figure]

The scientific approach is sound. The work presented is substantial (several simulations have been done and analyzed) but the conclusions and discussions deserve more work before publication. Several conclusions written in the manuscript are expected and already known. The risk is that the impact of the publication to a broader scientific community remains limited unless the authors put the conclusions into a wider perspective. In particular, this would require to add a section dedicated to a scientific discussion including more references to previous works on the subject if possible.

Specific comments

The added value of the CHIMERE model to the analysis must be better explained. The main conclusion I hold is the impact of the vertical resolution. The vertical resolution is certainly a factor of uncertainty for the representation of the boundary layer in general (with or without fire) and would deserve a full sensitivity analysis. At a minimum, this limitation of both models should be discussed.

The introduction refers to the study of PM2.5 and PM10. The latest are little discussed.

The fact that only one ground station was available to validate the model near fires moderates confidence in the findings. The paucity of the in-situ data (which is not the fault of the authors) should be pointed out in the general conclusion.

Both models include wet and dry deposition but little information is provided on the approaches. Are dry and wet depositions sources of uncertainties for PM2.5 and AOD? Wet deposition might not be predominant during the studied period.

The conclusion on the three months period evaluation says that the simulations are improved when fire emissions are included: is this surprising? It seems to me that the conclusion was expected unless there are previous studies that have concluded otherwise?

"Surface PM2.5 are particularly sensitive to the injection heights" is also an expected conclusion. Can the study help to decide between a PBL mixing of fire emissions vs.

injection height above PBL?

The introduction of the PPMfine fraction remains a little obscure for me and would deserve a little more explanation. It is important to understand this variable in light of its significant contribution to the composition of PM2.5 and the uncertainties in its definition (expected overestimation). How is this missing part treated in the other models?

Technical corrections

Figure 2: I assume that these are emissions calculated by APIFLAMME ?

Section 2.1: how are I/S/L VOCs represented in CHIMERE?

Table 2: I am confused between the Table marks and the comment above concerning the configuration of the CHIMERE model. The comment refers to simulation without I/S-VOCs and with fires but the table for the CHIMERE-ref shows a "Yes" for I/S-VOCs Change PB to PBL.

Figure 3 & 4: Legend: add "surface PM2.5"

―――――――――――――――――――

---

## Author Comment (AC1) · 16 Sep 2018

The comment was uploaded in the form of a supplement:
https://www.atmos-chem-phys-discuss.net/acp-2018-309/acp-2018-309-AC1-
supplement.pdf

---

## Author Response (AR1)

[revised manuscript text omitted]

The authors wish to thank the anonymous referee for the very helpful comments and corrections. All corrections have been included in this new version. A response to the general and specific comments is provided below (in blue).

**General comments**

*The main general comment for the manuscript is that several of the results presented can be considered as tentative since they are provided for and refer to a single site (GR0039A).*

We agree that the lack of available surface data close to the fire region is a limitation to our model evaluations. That is why an evaluation is also done using satellite observations.
A rural background AIRBASE station is located close to the fire region and discussed in the specific comments (comment 5). The station GR0035A is located in the same area (Athens) in Greece as the station GR0039A, already included. For GR0035A, observations show significantly higher background PM2.5 levels with large variability due to local influences. We consider that this specific station is probably strongly influenced by local emissions and decided not to add additional figures that may be confusing (the model resolution cannot resolve local influences). However, following to reviewer's comment, we added it in the revised version since it also shows the high spatial variability of the transported fire plumes.
In addition to observations of surface concentrations, the remote sensing observations from MODIS and AERONET are considered. Even if comparison to remote sensing adds another level of uncertainty (calculation of optical properties, vertically integrated datasets), it improves the representativity of the model validation.
A sentence on the problem of data availability for the evaluation of the composition and impact of fire plumes has been added to the conclusions (page 24, lines 4-11 in the revised paper): "The ability of Polyphemus/Polair3D and CHIMERE to simulate regional surface PM2.5 concentrations and optical properties (particularly AOD) was evaluated based on comparison to available measurements (8 AIRBASE stations and 6 AERONET stations). Only two out of the 8 AIRBASE stations (GR0039A and GR0035A in Greece) and 3 out of 6 AERONET stations (Lecce University in Italy, Blida in Algeria and Bucharest in Romania) are used because they are they are the most affected by wildfires (stations where fire contribution is higher than 10%). The lack of surface observations strongly limits this evaluation but it is partly complemented by comparisons to MODIS satellite-based observations of AOD.
Comparisons to surface and remote sensing observations show that the models can simulate enhancements of a good order of magnitude and +/- 1 day uncertainty in the timing."

*In addition, in the manuscript more results are presented for the first summer period wildfire event (20-31 July 2007) although the maximum of emissions is during the second summer period wildfire event (24-30 August 2007) as shown in Figure 2.*

More Figures and results are added in the appendix of the revised paper for the second wildfire event (24-30 August 2007). Figures of the PM2.5 composition in MedReg2 (most affected during that time period) are added for Polyphemus simulations (Cf. Specific comment 6). The fire contribution to PM2.5 composition is almost the same for both events. Figures of the mean total AOD at 550nm from MODIS, for Poly-ref and CHIMERE-ref during the second fire event also shows similar conclusions as during the first fire event.

**Specific comments**

1. **Page 6, lines 20-22:** Instead of primary aerosol oxidation, are you probably referring to the oxidation of I/S/L-VOCs by OH in the gas phase?

The sentence (page 6, lines 20-22): "Each primary aerosol undergoes one OH-oxidation reaction in the gas phase with ..." is replaced by "Each primary I/S/L-VOC undergoes one OH oxidation reaction in the gas phase with ..."

2. **Page 8, lines 8-9:** Please use a reference and explain why POA emissions taken into account are modeled as LVOCs.

In page 8, POA emissions are modeled as LVOC in a sensitivity study. This aims at studying the impact of ignoring I/S-VOC emissions in the gas phase and ignoring the semi-volatile properties of POA. The sentence (page8, lines 8-9) : "fire emissions in the PBL but without I/S-VOCs" is modified to "fire emissions in the PBL but without I/S-VOC emissions in the gas phase. In this sensitivity study, the semi-volatile properties of POA are not considered, and POA emissions are modeled as LVOCs".

3. **Pages 11-12, Section 3.3**: The overall performance of CHIMERE model is shown to be slightly better than Polyphemus model considering also the statistics in Tables 3 and 4. Under this view, you have to present results of CHIMERE model in Figures 3, 4 and 11 additional to those of Polyphemus model for comparison.

It is not easy to determine which model is better in terms of statistics, as the performance varies with the statistics considered. For PM2.5 statistics, the overall performance of CHIMERE is slightly better for bias, but it is slightly worse for correlation. In terms of errors, CHIMERE is slightly better for PM2.5, but Polyphemus is slightly better for AERONET AOD.
However, figures for CHIMERE (Figure A.1, Figure A.2, Figure A.3) are added in the appendix A, C and G respectively, in the revised paper. Conclusions for CHIMERE are similar to those for Polyphemus.

**Figure A.1 is added to the appendix A**

[Figure]

Figure A.1: " Daily mean PM2.5 and AOD at 550 nm from the simulation CHIMERE-ref averaged over the summer of 2007. The 8 AIRBASE and 6 AERONET stations, used in this work, are represented here in blue dots.

The sentence "Figure B1 in Appendix B shows that conclusions for CHIMERE are similar to those for Polyphemus." is added to the revised paper (page 11, line 26).

**Figure A.2 is added to the appendix C:**

[Figure]

Figure A.2: "Contribution from fires to the total surface PM2.5 as the difference between reference simulations and simulations without fires for CHIMERE during the first (left panel) and second (right panel) fire events."

The sentence "Figure C1 in Appendix C shows that the contribution of fires for CHIMERE are similar to that for Polyphemus." is added to the revised paper (page 13, line 5).

**Figure A.3  is added to the appendix G:**

[Figure]

Figure A.3: PM2.5 exceedance (days) from fires  simulated by CHIMERE (difference between the Poly-ref and Poly-Nofires)  during the summer 2007 (from 15 July to 30 August 2007)."

The sentence "Figure G1 in Appendix G shows that conclusions for CHIMERE are similar to those for Polyphemus." is added  to  the revised paper (page 25, lines 22-23).

4. **Pages 12, Figure 3:** Presenting similar maps using the data of MODIS is necessary to support better the comments in the manuscript about Figure 3.

Comparisons to MODIS are addressed in the response to the specific comment 8.

5. **Page 13, Section 3.4.1**: Figure 4 reveals several parts of the Mediterranean area being affected by the wildfire plumes for example Central and South Italy, Sicily, Bulgaria, almost the whole country of Greece etc. In the manuscript you are referring to only 2 stations being affected by the wildfire emissions during the first event and 3 stations during the second one. Aren't there additional AIRBASE or AERONET stations that could be used to support your analysis? (For example GR0035A is also a background suburban station where PM2.5 concentrations are being measured; is there a reason for not using measured data from this station?). Is the performance of the model good also for the remaining areas that are not affected by the fires?

*5.1) aren't there additional AIRBASE or AERONET stations that could be used to support your analysis?*

Measurement stations have been selected using the contribution from fires. Only stations with relative contribution higher than 10% are kept. In addition, for surface concentration observations, only background suburban and background rural stations with available data during the fire events are chosen in this study (Figure 3 left panel in the paper).

In the revised paper page 15 line 2, the sentences: "Figure 6 shows time series of daily observed and simulated aerosols at stations with significant impact from fires ..." are replaced by: " Figure 3.5

shows times series of daily observed and simulated aerosols at background suburban and background rural stations with available data during fire events and with a fire contribution higher than 10% ..."

As already explained in the answer to the general comments, another station in Greece (GR0035A) meets this criteria, close to the station already included and discussed in detail.

In the revised paper, we added in Figure 2.5 the time series of daily mean surface PM2.5 at that station (GR0035A). Although the temporal tendencies of the simulated PM2.5 concentrations at GR0035A are consistent with the observations, background PM2.5 levels are strongly underestimated (Bias= -85%). The observed background levels are significantly higher than those at the GR0039A station. This suggests the influence of local emissions that are missing in the model or that the resolution of the simulations does not allow to represent. This was the reason why the GR0035A was not initially considered in the paper.

However, following the reviewer's suggestion to include more surface sites, we have chosen to include it in the revised version because it is a good illustration of the high spatial variability of the transported fire plume which can not be captured in regional CTM simulations.

At the GR0035A station, the second observed peak during the second fire event is more than twice higher than the observed peak at the GR0039A and is measured one day earlier. In the simulations, the enhancement due to fires is similar in shape and magnitude, clearly highlighting the difficulty to simulate the exact temporal variability of emissions and transport of fire plumes in CTMs. Moreover, the station GR0035A is probably affected by a dust episode during the second fire event. Similar conclusions are found for AOD at the station Thessaloniki (near the station GR0035A), which seems to be also affected by dust especially during the second fire event. The observed value of Angstrom exponent at Thessaloniki station is lower on the 25$^{th}$ August ($\alpha=0.95$) suggesting that a large fraction of coarse mode particles (probably dust transport). Here, we didn't include the Thessaloniki station due to the lack of available AOD measurements during the first fire event where the fire contribution is more pronounced.

The sentences are added to page 17 line 11 in the revised paper: "At the GR0035A station, the temporal tendencies of the simulated PM2.5 concentrations are consistent with the observations. However the first PM2.5 peak is underestimated since PM2.5 levels are strongly underestimated (bias= -85%). The observed background levels are significantly higher than those at GR0035A station. This suggests the influence of local emission that are missing in the model or limitations due to the coarse model resolutions which can not represent this station."

and we added also these sentences to page 17 line 26 in the revised paper: "At the GR0035A station, the second observed peak during the second fire event is more than twice higher than the observed peak at the GR0039A and is measured one day earlier. In the simulations, the enhancement due to fires is similar in shape and magnitude, clearly highlighting the difficulty to simulate the exact temporal variability of emissions and transport of fire plumes in CTMs. Moreover, the station GR0035A is probably affected by a dust episode during the second fire event."

[Figure]

Figure 6: Times series from 15 July to 30 August of daily mean surface PM2.5 concentrations at the AIRBASE station GR0035A.

**5.2) Performance of the models (CHIMERE and Polyphemus) for areas which are not affected by fires:**

Table C.1: Statistics of models-to-measurements comparisons for the mean daily PM2.5 from the reference Polyphemus simulation (from 15 July to 30 August 2007).

| Station | localization | Mean observed PM2.5 | Mean simulated PM2.5 | Correlation (%) | MFB (%) | MFE(%) |
|---------|-------------|---------------------|----------------------|-----------------|---------|--------|
| FR12021 | Toulouse-France | 10.48 | 6.44 | 85.5 | -48 | 48 |
| ES0010R | Spain | 9.54 | 7.95 | 59.4 | -22 | 36 |
| FR15043 | Grenoble-France | 10.57 | 8.28 | 75.8 | -26 | 31 |
| FR33101 | Chambery-France | 6.84 | 8.56 | 87.3 | 22 | 28 |
| FR03043 | Marseille-France | 14.93 | 10.16 | 92 | -43 | 43 |
| FR33102 | Chambery Le haut-France | 9.74 | 8.37 | 86 | -17 | 23 |
| IT0495A | Italy | 17.39 | 8.19 | 83.4 | -76 | 76 |

Table C.2: Statistics of models-to-measurements comparisons for the mean daily PM2.5 from the reference CHIMERE simulation (from 15 July to 30 August 2007).

| Station | localization | Mean observed PM2.5 | Mean simulated PM2.5 | Correlation (%) | MFB (%) | MFE(%) |
|---------|--------------|---------------------|----------------------|-----------------|---------|--------|
| FR12021 | Toulouse-France | 10.48 | 8.80 | 76.6 | -22 | 32 |
| ES0010R | Spain | 9.54 | 8.41 | 56.1 | -13 | 30 |
| FR15043 | Grenoble-France | 10.57 | 10.44 | 73.9 | -3 | 32 |
| FR33101 | Chambery-France | 6.84 | 10.47 | 67.6 | 40 | 44 |
| FR03043 | Marseille-France | 14.93 | 10.71 | 85 | -39 | 41 |
| FR33102 | Chambery Le haut-France | 9.74 | 10.19 | 61.3 | 1 | 35 |
| IT0495A | Italy | 17.39 | 10.78 | 67 | -49 | 51 |

Regarding the statistical evaluation of the models, the mean simulated PM2.5 at the different stations which are not affected by fires is in good agreement with the observations, except for IT0495A where PM2.5 are underestimated compared with surface PM2.5 measurements.
The model performance is always met and the model goal is met only for the model errors MFEs. The models-to-measurements correlations range between 59.9 and 92% for all the stations.

Generally, the performance of the two models (Polyphemus and CHIMERE) is good for the areas which are not affected by fires, except at IT0495A where PM2.5 background levels in the simulations are underestimated most probably due to dust.

Since this paper focuses on the evaluation of models performance near the most affected stations by fires, we choose not to include the model performance at the stations which are not affected by fires in the revised paper.

6. **Page 14-15, Section 3.4.2:** The results presented in this section refer only to a single station and only to the fire event of the first period. Under this view, they are of very limited reference and cannot be used to characterize the aerosol composition in the Mediterranean areas affected by the fire plumes. Similar results with those presented in Figure 5 have to be presented for different stations and wider areas affected by the fire emissions and separately for both fire peak events in order possible differences to be identified.

Figure 5 page 15 is replaced by Figure 3.3 and Figure 3.4 in the revised paper.
Figure 3.3 and Figure 3.4 show the composition of surface PM2.5 concentrations for two wider areas which are the subregions: MedReg1 (for the first fire event) and MedReg2 (for the second fire event). These two subregions are the areas most affected by fires (high fire emissions (Figure 2)) especially during the first fire event for Medreg1 and during the second fire event for MedReg2.

[Figure]

Figure 3.3: Composition of surface PM2.5 concentration over MedReg1 during the first fire event (upper left panel, simulation Poly-Nofires). Composition of surface PM2.5 concentration due to fires (upper right panel: simulation Poly-ref; lower left panel: simulation Poly-NoI/S-VOCs; lower right panel: simulation Poly-NoPPM).

[Figure]

Figure 3.4: Composition of surface PM2.5 concentration over MedReg2 during the second fire event (upper left panel, simulation Poly-Nofires). Composition of surface PM2.5 concentration due to fires (upper right panel: simulation Poly-ref; lower left panel: simulation Poly-NoI/S/L-VOCs; lower right panel: simulation Poly-NoPPM).

Similar composition is found for surface PM2.5 concentrations over two wider areas: MedReg2 during the second fire event and over MedReg1 during the first fire event for both models (CHIMERE and Polyphemus).

During the first fire event over MedReg1, organic aerosols contribute significantly to surface PM2.5 concentrations. Their contribution is larger than those of other species such as inorganics (11%) PPMfine (dust) (15%) and black carbon (5%). When we consider fire emissions, we found that organic aerosols which are mostly composed of POA and SOA from I/S/L-VOCs, are predominant in the contribution of fires (between 47% and 85% of the contribution). Lower contribution are found for inorganics (8% to 13%) and black carbon (3% to 6%).

During the second fire event over MedReg2, PM2.5 composition in Poly-Nofire is similar to PM2.5 composition during the first fire event over MedReg1. The same conclusions are found as during the first fire event: the organic aerosols are the most important contributor in the composition of PM2.5 from fires (46% to 81%). Lower contributions are found for inorganics (9 to 12%) and black carbon (5 to 6%).

The sentences (page 14 line 2-23) are corrected as followed: "The composition of surface PM2.5 concentrations simulated during the first fire event over MedReg1 and during the second one over MedReg2 are shown in Figures 3.3 and 3.4 respectively. These two subregions are the areas most affected by fires (high fire emissions (Figure 2)) especially during the first fire event for Medreg1 and during the second fire event for MedReg2.

The upper left panel shows the composition of surface PM2.5 concentrations for the simulation without fire Poly-NoFires (background surface PM2.5 concentrations), while the upper right panel shows the composition of surface PM2.5 concentrations due to fires (differences between the simulations Poly-ref and Poly-Nofires). If wildfires are not taken into account (simulation without fire) and during the first fire event over MedReg1, organic and inorganic aerosols contribute equally (42.6%, 40.5%) to the surface PM2.5 concentrations. The contribution of PPMfine (dust), black carbon are lower (15%, 1%). As noted by Chrit et al. (2017), most of summer organic aerosols are from biogenic sources in this region.

If wildfires are not taken into account during the second fire event over MedReg2, inorganics and PPMfine are the predominant component in the composition of PM2.5 concentrations (56.5% and 27.9%). Lower contributions are simulated for black carbon (1.2%) and organic aerosol (14.3%).

Figures 3.3 and 3.4 also show the composition of surface PM2.5 concentrations due to fires for the simulations Poly-ref (differences between the simulations Poly-ref and Poly-Nofires), Poly-NoI/S-VOCs (differences between the simulations Poly-NoI/S-VOCs and Poly-Nofires) and Poly-NoPPM (differences between the simulations Poly-NoPPM and Poly-Nofires). During the first fire event over MedReg1, organic aerosol is predominant in the contribution of fires (between 47% and 85% of the contribution). Organic aerosol is mostly composed of POA and SOA from I/S/L-VOCs (46% to 80%). Note that POA and SOA from L-VOCs (low volatile organic compounds) are important even in the simulation when I/S-VOCs are not taken into account in fire emissions (46%), because POA are then assigned to L-VOCs. The contribution from inorganics (8% to 13%) and black carbon (3% to 6%) are low. During the second fire event over MedReg2, similar PM2.5 composition is found. Organic aerosol (mainly the POA and SOA from I/S/L-VOCs) is the most important component contributing to PM2.5 from fires (between 46% and 81% of the contribution). The contribution from inorganics ( 9% to 12%) and black carbon (5 to 6%) are lower.

Similar PM2.5 composition is found not only during the second fire event over MedReg2 but also in the concentrations simulated by CHIMERE (not shown here). Surface PM2.5 concentrations from fire simulated by CHIMERE are composed in the first and second fire events mainly of organic aerosol, mostly composed of primary organic carbon (OCAR) (46%) which corresponds to I/S/L-VOCs in Polyphemus, and of PPMfine (39%). The contributions from inorganics (9%), black carbon (5.2%) and SOA from VOCs (2.7%) are low."

7. **Page 16, lines 4-8:** Comments for Figure 8 is better to be written at the end of the section 3.4.3 after the comments for Figures 6 and 7.

Comments for Figure 8 are moved at the end of the section 3.4.3 (page 18 line 12 in the revised paper).

8. **Page 16, Figure 8:** Please present and comment on similar maps based on the models and MODIS results for the second fire event in August.

Figure D.1 is added to Appendix D and E.1 are added to Appendix E in the revised paper.

[Figure]

Figure D.1: "Mean total AOD (at 550 nm) from MODIS/AQUA, Poly-Ref and CHIMERE-ref during the second fire event (24-30 August 2007)."

These sentences are added in the revised paper (page 18 line 16) (after moving the comments for Figure 8 at the end of the section 3.43 page 18 line 12): " During the second fire event in Greece, the simulated AOD for Poly-ref and for CHIMERE-ref is about 1 and 0.9 respectively as shown in Figure D.1 in Appendix D. The observed AOD can reach 0.9. However, both models overestimate AOD values in the fire plume (reaching ~0.98 for Poly-ref and 0.88 for CHIMERE-ref against 0.7-0.8 in Greece and 0.5 from MODIS. The fire plume is less pronounced in observations than in simulations.

Day-to-day comparisons for four selected days (24, 25, 27 and 29 August 2007) are shown in Figure E.1 in Appendix E. The simulated AOD is consistent with the observations in terms of localization and general transport pathways. However, the simulated AOD is much higher in the Greek fires' plume compared with MODIS observations during the peak of emissions (25-29 August). This probably reflects too low temporal variability in the emissions. In the simulations, emissions are assumed constant during the day but comparisons suggest shorter temporal variability. This is also apparent in the time series of Figure 3.6 in the paper, over region MedReg2: the peak for the second fire event is twice longer in the simulations (double peak starting on the 25 August 2007) than in the observations. This peak over two days instead of one in the simulations suggests an overestimate of emissions during this event which is also observed with respect to surface observations in Greece. The first peak corresponds better to observations. This shows that uncertainties are not only related to total emissions but also to their temporal variability and the associated transport pathways."

[Figure]

Figure E.1: "Total AOD (at 550 nm) from MODIS/AQUA and the corresponding  Polyphemus AOD (Poly-Ref) and  CHIMERE AOD (CHIMERE-ref)  for four selected days (24, 25, 27 and 29 August 2007)."

**9. Page 23, Section 4.1:** In lines 10-12, it is commented that according to the Polyphemus model results the number of days, when the WHO air quality limit for the daily PM2.5 values is exceeded in GR0039A station, is 7 while the corresponding number based on measurements is 14. This corresponds to a -50% bias by the model. Is it acceptable? No data are provided for additional stations in the study area rendering the results presenting in this section very tentative. Further evidence is necessary to prove the acceptable performance of Poly-ref runs (and CHIMERE-ref runs, please see na previous comment) in the estimation of the PM2.5 air quality exceedences supporting better the necessity for the maps in the Figure 11 to be presented.

Table F.1: Modeled and observed PM2.5 exceedances at each AIRBASE stations. Values between brackets correspond to modeled PM2.5 exceedances for simulations without fire emissions (Poly-Nofires and CHIMERE-Nofires).

| Airbase station | Observed PM2.5 exceedances | | | Modeled PM2.5 exceedances **Poly-ref** | | | Modeled PM2.5 exceedances **CHIMERE-ref** | | |
|---|---|---|---|---|---|---|---|---|---|
| | **Whole period** | **First fire event** | **Second fire event** | **Whole period** | **First fire event** | **Second fire event** | **Whole period** | **First fire event** | **Second fire event** |
| GR0039A | 14 | 5 | 5 | 7 (0) | 3 (0) | 4 (0) | 9 (2) | 3 (1) | 5 (0) |
| GR0035A | 37 | 12 | 7 | 7 (0) | 3 (0) | 4 (0) | 9 (2) | 3 (1) | 5 (0) |
| IT0459A | 4 | 2 | 2 | 0 (0) | 0 (0) | 0 (0) | 0 (0) | 0 (0) | 0 (0) |
| FR12021 | 0 | 0 | 0 | 0 (0) | 0 (0) | 0 (0) | 2 (2) | 0 (0) | 2 (2) |
| ES0010R | 0 | 0 | 0 | 1 (1) | 0 (0) | 1 (1) | 2 (2) | 0 (0) | 2 (2) |
| FR15043 | 0 | 0 | 0 | 0 (0) | 0 (0) | 0 (0) | 4 (4) | 0 (0) | 4 (4) |
| FR33101 | 0 | 0 | 0 | 0 (0) | 0 (0) | 0 (0) | 3 (3) | 0 (0) | 3 (3) |
| FR03043 | 5 | 0 | 3 | 2 (2) | 0 (0) | 2 (2) | 4 (4) | 0 (0) | 4 (4) |
| FR33102 | 0 | 0 | 0 | 0 (0) | 0 (0) | 0 (0) | 3 (3) | 0 (0) | 3 (3) |

Table F.1 is added to the revised paper in the appendix F.

The sentences in page 23, lines 10-12: "The number of exceedances predicted by the model is first compared to exceedances observed by AIRBASE (only one station is available). During the summer 2007, 14 PM2.5 exceedances were observed and only 7 are simulated at GR0039A by Poly-ref and 9 by CHIMERE-ref." are replaced by "Table F1 in Appendix F presents the modeled and observed PM2.5 exceedances at each AIRBASE stations during the whole summer 2007 and during fire events. Generally, the models (Polyphemus and CHIMERE) underestimates PM2.5 air quality exceedances because the horizontal resolution used here does not allow the representation of local pollution (especially for the station GR0035A). Better performance is observed during fire events than the whole period. Near fire regions, at the stations GR0039A, the number of exceedance is well-modeled, in spite of the slight underestimation during both fire events compared to the observed PM2.5 exceedances. However, at the GR0035A station, the underestimation of PM2.5 exceedances is sharp. This can be explained by the strong underestimation of the background PM2.5 levels as shown in Figure 3.5. Far from fire regions, the PM2.5 exceedances modeled by Polyphemus are in a good agreement with the observed ones especially during the two fire events. The simulated PM2.5 exceedances by CHIMERE in the fire plume are overestimated at some stations ."

**10. Page 23, Figure 11:** Figure 11 is not clearly explained in the manuscript. What is presented in Figure 11 (left panel)? Is it the additional days with exceedences due to fires (i.e. (days with exceedences Poly-ref) minus (days with exceedences Poly-Nofires))?

This sentence (page 23 lines 13-15): "Figure 11 shows the simulated number of days concerned by a PM2.5 exceedance from fires (difference of PM2.5 exceedance days between the Poly-ref and Poly-Nofires). Most are concentrated around fire sources, mainly in Balkan (30 days)." is replaced by "Figure 4.3 shows the additional days with PM2.5 exceedances due to fires simulated by Polyphemus (difference of PM2.5 exceedance days between the Poly-ref and Poly-Nofires). Most are concentrated around fire sources, mainly in Balkan (30 days)."

Figure 11 's caption "PM2.5 exceedance (days) from fires simulated by Polyphemus (difference between the Poly-ref and Poly-Nofires) (upper left panel) and simulated by CHIMERE (difference between the Poly-ref and Poly-Nofires) (upper right panel) and the maximum dispersion (σ) related to fire emissions for PM2.5 exceedances days (lower left panel) during the summer 2007 (from 15 July to 30 August 2007)." is replaced by "The additional days with PM2.5 exceedances due to fires simulated by Polyphemus (difference between the Poly-ref and Poly-Nofires) and the maximum dispersion (σ) related to fire emissions for additional days of PM2.5 exceedances due to fires (lower left panel) during the summer 2007 (from 15 July to 30 August 2007).

**11. Page 23, lines 16-18:** It is commented by the authors that PM2.5 concentrations are composed mainly of organic matter. However adding the percentages in the parentheses (i.e. 7.3% for POA and 19.1% for SOA), a percentage total contribution of 26.1% is estimated. Is this contribution to PM2.5 the main? Please clarify.

The sentence (page 23, lines 16-18): "During the fire periods, surface PM2.5 concentrations simulated in Poly-ref in MedReg1 and MedReg2 sub-regions are composed mainly of primary organic matter (7.3% with 92% due to fire) and secondary organic aerosol (19.1% with 78.9% due to fires)." is replaced by "During the fire periods, surface PM2.5 concentrations simulated in Poly-ref in MedReg1 and MedReg2 subregions are composed mainly of OM (54% with 61.6% due to fire for MedReg1 and 52.3% with 60.1% due to fires for MedReg2), and inorganics (27% with 8% due to fires for MedReg1 and 15% with 9% due to fires for MedReg2) ."

P.S.: *During each fire event, over each subregion, the percentage (P) presenting the contribution of each component (X) from fire, is calculated as follows :* $P\% = [X]_{poly\text{-}ref} - [X]_{poly\text{-}Nofire} / [PM2.5]_{Poly\text{-}ref.}$

[Figure]

[Figure]

Figure: composition of surface concentrations over MedReg1 during the first fire event and MedReg2 during the second fire event for simulation Poly-Ref.

**12. Page 24, lines 9-10:** The percentages have been estimated for one site and for one fire event and have a very limited reference (spatial and temporal). Please see a previous comment of mine so as to update the conclusions.

We replaced the percentages given for one site and for one fire event by percentages for wider areas during the two fire events : MedReg1 during the first fire event and MedReg2 during the second fire event.
The sentence in page 24, lines 9-10: "Near fires, PM2.5 is mostly composed of organic aerosol (52% to 87%), with a strong contribution from I/S/L-VOCs (62% to 84%)." is replaced by "Near fires, PM2.5 is mostly composed of organic aerosol (47% to 85%), with a strong contribution from I/S/L-VOCs (46% to 80%)."

**13. Page 24, lines 10-12:** Which Figure or Table supports these high correlation values and where in the manuscript are these results commented? Is it Figure 7?

Figure 7 supports these high correlation values. These results are commented (page 15 lines 6-8).

**14. Page 24, lines 14-16:** This conclusion is not straightforward and requires further clarifications not only in conclusions but also in the manuscript. In addition, it is not clear how overestimations in simulated results may indicate missing emissions.

Indeed, this sentence is not clear. The sentence (page 24, lines 14-16): "The overestimation is very low (10%) in the simulation Poly-NoPPM, where I/S-VOCs are taken into account, but unidentified primary particles emitted from biomass burning are not." is replaced in the revised paper by "Since unidentified primary particles (PPMfine) emitted from biomass burning are not considered in Poly-NoPPM, the simulated AOD values are closer to MODIS observations. This suggests that PPMfine could correspond to I/S-VOCs."

The authors wish to thank the anonymous referee for the very helpful comments and corrections. All corrections have been included in this new version. A response to the general and specific comments is provided below (in blue).

**# General comments:**

*A) This paper tries to evaluate model estimates of fire-related PM concentrations through evaluation with data from surface networks and satellite remote sensing. However, I think this work is flawed due to the decision to use only PM measurements. For some reason they do not make use of the rather extensive measurements which are available (Airbase,EMEP, or even AMS data) of sulfate, nitrate, ammonium, or other inorganic components. This is a serious problem I think. For example, Fountoukis et al (2011) suggested significant contributions of sulfate, nitrate and ammonium to PM1 mass, and backed up their analysis with comparison to measurements. Van Damme et al. (2014) showed enormous levels of NH3 from biomass burning, which presumably caused high PM2.5 also.*

Figure 1 shows the PM2.5 composition for simulation "Poly-Nofires" over the Euro-Mediterranean region and during the summer 2007. Inorganics, mainly sulfate (23.5%), sea salt (23.6%) and ammonium (7.6%), contribute highly to PM2.5 composition. Similar results are found in Fountoukis et al. (2011) who showed also that sulfate, sea salt and ammonium contribute highly to the PM1 mass (30%, 14% and 13% respectively) over Europe during May 2008.
Several other studies evaluated the performance of Polyphemus to simulate sulfate, ammonium and nitrate concentrations (Sartelet et al., 2007; Zhang et al., 2013; Chrit et al., 2018). Chrit et al., (2018) evaluated the model Polyphemus over the western Mediterranean region during the ChArMEx campaign, by comparison to in-situ aerosol measurements performed during three consecutive summers (2012, 2013 and 2014). According to Chrit et al. (2018), sulfate and ammonium concentrations satisfy the performance and goal criteria, while sodium satisfies only the performance criteria. Chrit et al. (2018) found that chloride and nitrate were both underestimated compared to observations. They explained that this underestimation is probably caused by uncertainties in the measurements and difficulties in representing the gas-particle partitioning. Sartelet et al. (2007) showed that the concentrations of inorganics modeled with Polyphemus over Europe are consistent with observations (EMEP and AIRBASE). Model performance criteria are met for sulfate, nitrate and ammonium. Zhang et al. (2013) used Polyphemus to simulate air quality in July 2001 over western Europe. They evaluate the model's performance using surface measurements from EMEP and AIRBASE stations for inorganics. The simulated concentrations of inorganic PM are in a good agreement with EMEP observations, although sulfate and nitrate concentrations are underestimated.

The formation of inorganics because of wildfires is found to be low compared to the formation of organics. However, our simulation takes into account emissions of inorganic precursors such as ammoniac (NH3). Several studies (R'Honi et al.,2013; Van Damme et al.,2014; Whitburn et al., 2017), show that large emissions of NH3 are released by biomass burning. Whitburn et al. (2017)

studied the enhancement ratios NH3/CO for biomass burning emissions in the tropics using observations from the IASI satellite based instrument. They found a significant variability due to fire contribution. According to the Whitburn et al. (2017), the emission ratios NH3/CO in the tropics derived from IASI observations (as in Van Damme et al., 2014) are rather on the lower end of those reported in Akagi et al. (2011) that are used here.

If fire emissions are important for the regional budget of inorganics, more observations are required to provide emissions values of NH3 and concentrations of inorganics should be evaluated close to fire regions.

During the summer 2007, AMS data are not available. The PM data from AIRBASE and EMEP are scarce in our domain. We only have a few PM2.5 stations and only 3 stations (far from the studied fire regions: 2 in Spain and 1 in France) for inorganic concentrations. Evaluation with measurements is therefore not possible. However, we looked more closely at the contribution of inorganics to the simulated concentrations.

[Figure]

Figure 1: Composition of surface PM2.5 (simulation Poly-No-fires) over the Euro-Mediterranean region during summer 2007

[Figure]

Figure 2: Composition of surface PM2.5 concentration due to fires (left panel: simulation Poly-ref over MedReg1 during the first fire event and right panel: simulation Poly-ref over MedReg2 during the second fire event).

Poly-Nofires

(Poly-ref - Poly-Nofires)

Poly-Nofires

(Poly-ref - Poly-Nofires)

Figure 3: Composition of surface inorganic concentrations over Medreg1 during the first fire event (upper left panel, simulation Poly-nofires) and over MedReg2 during the second fire event (lower left panel, simulation Poly-nofires). Composition of surface inorganics due to fires over Medreg1 during the first fire event due to fires (upper right panel: simulation Poly-ref) and over MedReg 2 during the second fire event (lower right panel: simulation Poly-ref).

The contribution of inorganics is low compared to the contribution of organics. Figure 2 and Figure 3 show the composition of surface PM2.5 concentrations due to fires for simulation Poly-ref and the composition of surface inorganic concentrations respectively over MedReg1 for the first fire event and over MedReg2 for the second fire event.
The contribution of inorganics from fires varies between 8.3 and 9.3%. Focusing on the contribution from fires to inorganics, sulfate, ammonium and nitrate are the predominant components: between 55.7% and 67.6% for sulfate, between 26.8 and 38.7% for ammonium and 5.6 to 13.6% for nitrate.

These sentences are added to page 14 line 24 of the revised paper: "In our study, inorganics (mainly sulfate, sea salt and ammonium) contribute highly to PM2.5 composition, if fire emissions are not considered. Similar results are found in Fountoukis et al. (2011) who showed the high contribution of sulfate, sea salt and ammonium to PM over Europe during May 2008. However, when fire emissions are taken into account, the contribution of inorganics from fires is lower than the contribution of organics (8 to 9% of inorganics against 40 to 80% of organics). Focusing on the contribution from fires, sulfate, ammonium and nitrate are the predominant components of inorganics from fires: between 55.7% and 67.6% for sulfate, between 26.8 and 38.7% for ammonium and 5.6 to 13.6% for nitrate."

*B) Further, on p10 they claim that by focusing on PM2.5 they avoid contributions from dust, but this simply isn't the case. The low-diameter tail of coarse-mode dust (and sea-salt) contributes of course to PM2.5, and measurements in southern Europe have shown that this can be a significant fraction even over long-periods (Putaud et al., 2010). (This same publication also demonstrates clearly the importance of non-organic compounds in contributing to PM2.5.)*
*I see no reason to believe that changes in model performance attributed here to various assumptions on OA emissions could not be masked or mistaken for problems with inorganic pollutants. As well as attending to the points raised below, I would encourage the authors to do a thorough analysis of the model performance for all the major compounds, and across many sites. I am afraid I cannot recommend this paper for publication without major revision.*

Indeed, the low-diameter tail of coarse-mode dust (and sea-salt) contributes of course to PM10, but we choose not to consider PM10 in this paper (and therefore focus on PM2.5) in order to reduce the uncertainties related to dust emissions as much as possible.

Sentences page 10 line 5-7 "PM10 concentrations in the Mediterranean area are strongly affected by dust, which are difficult to simulate due to their sporadic nature and their main sources are located out of the model domain. Since dust is not in the scope of this paper, the analysis focuses on PM$_{2.5}$." are replaced by " PM10 concentrations in the Mediterranean area are strongly affected by dust, which are difficult to simulate due to their sporadic nature and the fact that their main sources are located out of the model domain. In order to evaluate more specifically the uncertainties associated with fine particles (largest contribution from fires), and to minimize the contribution from dust, the analysis focuses on PM$_{2.5}$ ."

**Other points:**

**0. The title is actually quite misleading. The paper only addresses some aspects of OM emissions, not emissions in general.**

The title : "Impact of wildfires on particulate matter in the Euro-Mediterranean in 2007: sensitivity to the parameterization of emissions in air quality models." is replaced by : "Impact of wildfires on particulate matter in the Euro-Mediterranean in 2007: sensitivity to some parameterizations of emissions in air quality models."

**1. The word dispersion, used in eg line 17 of the abstract, is usually associated with turbulent mixing in the air pollution context. Please use another term to make clear that here you mean statistical spread.**

The word "dispersion" used in the paper is replaced by the "statistical dispersion" to avoid any kind of confusion.

**2. p2, L20. Give reference for statements about anthropogenic origin.**

The sentence page 2, line 20-21 : "Although ignitions are mainly of anthropogenic origin (negligence, arson, agricultural practices) , fire spread depends on meteorological conditions." is replaced by: "Although ignitions are mainly of anthropogenic origin (negligence, arson, agricultural practices) according to San-Miguel-Aynanz et al. (2013) and the European Forest Fire Information System (EFFIS) of the European Joint Research Center (JRC), fire spread depends on meteorological conditions."

**3. p3, L8. Give reference for the volatility limits and definitions. Also, do you really mean from 1e4, or from 0.32e4 (see eg Table 4 of Murphy et al., 2014) ?**

The volatility borders are very uncertain, but in this work we followed the volatility limits given in Robinson et al. (2007). The reference is added in the revised paper (page 3, line 12).

Table 1 summarizes the main characteristics of the Polyphemus and CHIMERE simulations. All the Acronyms used in this Table are explained in the text (from page 5 lines 1-21 to page 6 lines1-23). The clarifications below are added in the revised paper.

**6-1 What is ECMWF, which version ?**

The sentence  page 5 line 20-21 "Both models (Polyphemus and CHIMERE) are driven by meteorological conditions simulated by the European Center for Medium-Range Weather Forecasts (ECMWF) model. " is replaced by "Both models (Polyphemus and CHIMERE) are driven by meteorological conditions simulated by the European Center for Medium-Range Weather Forecasts (ECMWF, ERA-Interim) model."

**6-2 What does 'from nesting simulations' mean?**

Boundary conditions are from a simulation undertaken using a large domain (0.5°x0.5°, horizontal resolution) covering Europe and North Africa (see Figure G).
The Figure G is added in the appendix  A in the revised paper.
The sentence page 6 line 3: " Simulations are undertaken using two nested domains." is replaced by " Simulations are undertaken using two nested domains (Figure A1 in Appendix A)."

[Figure]

Figure G: Simulation domains including one large domain (with a 0.5◦ ×0.5◦ horizontal resolution) and a smaller domain (at a 0.25◦ ×0.25◦ horizontal resolution) delimited by the dotted red box.

**6-3 Which version of MEGAN?**

The sentence page 6 line 9-10 "Biogenic emissions are calculated using the Model of Emissions of Gases and Aerosols from Nature (MEGAN) (Guenther et al., 2006)." is replaced by " Biogenic emissions are calculated using the Model of Emissions of Gases and Aerosols from Nature (MEGAN) with the standard MEGAN LHIV database MEGAN-L for Polyphemus and MEGAN v2.04 or CHIMERE (Guenther et al., 2006)."

**6-4 Which version of EMEP emissions?**

The sentences page 6 line 4-5 "Anthropogenic emissions are derived from the EMEP emissions inventory (European Monitoring and Evaluation Program, www.emep.int)." is replaced by "Anthropogenic emissions are derived from the EMEP emissions inventory for 2007. (European Monitoring and Evaluation Program,www.emep.int)"

**7. P6, L3. The nesting factor is here given as 2:1, but usually one uses an odd ratio to avoid interpolation errors (e.g.**
https://earthscience.stackexchange.com/questions/8067/why-is-wrf **most-often-configured-at-31-nesting-ratio). Have the authors checked if their results are robust?**

The recommendation to use an odd ratio applies to WRF simulations. However, this is not required by Polyphemus nesting. Indeed the interpolation is done properly in 2D in Polyphemus, and it is accurate because the vertical levels do not vary with time/space.

**8. p6, L9. Biogenic emissions of what? I assume isoprene and terpenes, but which?**

The sentence page 6 lines 9-10 " Biogenic emissions are calculated using the Model of Emissions of Gases and Aerosols from Nature (MEGAN) (Guenther et al., 2006)." is replaced by "Biogenic emissions of isoprene and terpenes (α-pinene, β-pinene, limonene and humulene) are calculated using the Model of Emissions of Gases and Aerosols from Nature (MEGAN) (Guenther et al., 2006)."

**9. p6, L17-24. The explanation of the VBS scheme used is unclear. Firstly, where do the numbers -0.04, 1.93 and 3.5 come from? They are not suggested by May et al. 2013, and not obvious to the reader. Secondly, L21 suggests that OH+POA leads to SOA compounds in the same VBS bin, but L22-23 suggests a reduction in volatility equivalent by a factor of 100. Which is it?**

In this work, we didn't use a VBS scheme. We used the one-step oxidation scheme developed by Couvidat et al. (2012) where the emission distribution is based on the fitting of the curve of dilution of diesel exhaust from Robinson et al. (2007). This emission distribution is approximately similar to the one measured by May et al. (2013a) for biomass burning (see Table A). According to Couvidat et al. (2012), I/S/L-VOCs emissions are assigned to 3 surrogates species: POAlP, POAmP and POAhP (for compounds of low, medium and high volatilities respectively), of saturation concentration $C^*$ : $\log_{10}(C^*) = -0.04, 1.93, 3.5$ respectively.

| C* (µg/m³) | Couvidat et al. (2012) emissions distribution | May et al. (2013a) emissions distribution for biomass burning |
|---|---|---|
| $10^{-0.04}$ | 0.25 | 0.20 0.20 |
| $10^{1.93}$ | 0.32 | 0.10 0.20 |
| $10^{3.5}$ | 0.43 | 0.4 |

Table A: the emission distribution of I/S/L-VOCs given by Couvidat et al. (2012) and May et al. (2013a)

The sentences in page 6, lines 17-24: "I/S/L-VOCs emissions are assigned to 3 surrogates species: POAlP, POAmP and POAhP (for compounds of low, medium and high volatilities respectively), of saturation concentration $C^{\pi}$ : log(C*) = -0.04, 1.93, 3.5 respectively. The volatility distribution at emissions of I/S/L-VOCs is detailed in Couvidat et al. (2012) (25%, 32%, and 43% of I/S/L-VOCs are assigned to POAlP, POAmP and POAhP respectively). It corresponds to the volatility distribution measured by May et al. (2013) for biomass burning aerosol emissions. Each primary aerosol undergoes one OH-oxidation reaction in the gas phase with a kinetic rate constant equal to $2.10-11$ molecule−1 .cm3 .s−1 , leading to the formation of secondary surrogate in the same volatility: SOAlP, SOAmP and SOAhP. The ageing of the primary aerosols reduces the volatility of the secondary product by a factor of 100 and increases the molecular weight by 40% (Couvidat et al., 2012). " are replaced by "I/S/L-VOCs emissions are assigned to 3 surrogates species: POAlP, POAmP and POAhP (for compounds of low, medium and high volatilities respectively), of saturation concentration C*: log(C*) = -0.04, 1.93, 3.5 respectively. The volatility distribution at emissions of I/S/L-VOCs is detailed in Couvidat et al. (2012) (25%, 32%, and 43% of I/S/L-VOCs are assigned to POAlP, POAmP and POAhP respectively). It corresponds to the volatility distribution measured by May et al. (2013a) for biomass burning aerosol emissions. Each primary aerosol undergoes one OH-oxidation reaction in the gas phase with a kinetic rate constant equal to $2.10^{-11}$ molecule$^{-1}$.cm$^3$ .s$^{-1}$, leading to the formation of secondary surrogates: SOAlP, SOAmP and SOAhP. The ageing of the primary aerosols reduces the volatility of the secondary product by a factor of 100 and increases the molecular weight by 40% (Couvidat et al., 2012)."

**10. p6, L17. The use of 'log' is ambiguous. In most VBS papers it means log10, but in other papers (especially mathematical) and common programming languages (eg python) it means the natural logarithm, log_e (or ln). I suggest using log10 to be clear.**

The sentence page 6, line 17 : "I/S/L-VOCs emissions are assigned to 3 surrogates species: POAlP, POAmP and POAhP (for compounds of low, medium and high volatilities respectively), of saturation concentration C* : log(C* ) = -0.04, 1.93, 3.5 respectively." is replaced by "I/S/L-VOCs emissions are assigned to 3 surrogates species: POAlP, POAmP and POAhP (for compounds of low, medium and high volatilities respectively), of saturation concentration C* : $\log_{10}$(C* ) = -0.04, 1.93, 3.5 respectively."

**11. p6, L33-34. Akagi provides emission factors for several ecosystems, but non are a close match to Mediterranean landscapes. And some of these emitted isoprene and terpenes, which also form SOA. Which emissions were in fact used?**

The sentence in page 6 line 33 "Emissions for each species are derived from the carbon emissions using the emissions factors from Akagi et al. (2011)." is replaced by "Emissions for each species are derived from the carbon emissions using the emissions factors from Akagi et al. (2011). These emission factors are provided in terms of g species per Kg Dry Matter (DM) burned (g.kg$^{-1}$) for all relevant species observed in biomass burning plumes and for different standard vegetation types that match to Mediterranean landscapes (chaparral, temperate forest, crop residue, pasture maintenance and savanna). The contribution of these vegetation types to the burned area detection over the Mediterranean region during the time period studied is 37.2% for temperate forest, 32.7% for savanna, 9.6 % for chaparral and 19.9% for crop residue."

The isoprene and terpenes (α-pinene, β-pinene, limonene and humulene) emissions from fires are considered in this work.

**12. Also, L33 suggests that only carbon emissions were used, but many studies suggest that fires are a major and sometimes huge source of gases including NH3 (eg van der Werf et al., 2017, Van Damme et al., 2014), and hence of inorganic PM. Were such emissions examined for their impact on PM? Were they checked?**

According to the Whitburn et al. (2017), the emission ratios NH3/CO in the tropics derived from IASI observations (as in Van Damme et al., 2014) were rather on the lower end of those reported in Akagi et al. (2011) (used here). Therefore, we believe we did not underestimate NH3 emissions. However, the impact on inorganic concentrations is low, as discussed previously in comment 1.

**13. p7, The notation PPMfine is very confusing. This is usually used to represent emissions of any primary particulate matter including OM, BC and inorganics. Am I right in saying that your PPMfine is PM2.5 - OC - BC - SIA? In which case I would expect most of be essentially OM-OC - is that what you mean? To further the confusion with this PPMfine notation, you introduce PPMcoarse which is more traditional, just PM10-PM2.5. So, PPMfine excludes OC, BC and SIA, and consists largely of O and H, whereas PPMcoarse consists of OM+BC+SIA. As I said, confusing.**

PPMfine corresponds to all the unidentified fine particles emitted by wildfires which are incorporated to consider the differences between PM2.5 emissions and the total of all PM included in specific species. Therefore, PPMfine corresponds to PM2.5-OM-BC-SIA.

This is why we believe that PPMfine includes actually OM and I/S/L-VOCs.

For clarity, the following sentences are added is added in page 7 lines 11-14: "PPMfine corresponds to all the unidentified fine particles emitted by wildfires which are incorporated to consider the differences between PM2.5 emissions and the total of all PM2.5 speciated emissions."

**14. p7, Figure 1. It is difficult to see the yellow fire labels on top of the yellow background. Change the colors.**

In the revised paper, the Figure 1 page 7 is replaced by the Figure A below.

[Figure]

Figure A: Map of the nested domain over the Mediterranean area with a spatial resolution of 0.25°×0.25°. The total organic carbon emissions(kg. (grid cell)−1 ) from fires during the summer of 2007 are presented. The sub-regions used in this study are also indicated in colored boxes: MedReg1 (Balkan + Eastern Europe), MedReg2 (Greece), MedReg3 (Italy) and MedReg4 (Algeria).

**15. p9. Table 2.**

**- The Poly-ref emissions are said to be introduced between 1km and the PBL. I assume you mean between ground level and min(PBL,1km)? If not, I have a serious problem with your base-case!**

Indeed, we meant that fire emissions in the simulation Poly-ref by are injected in min (PBL, 1km).

**- What is PB?**
We mean by PBL : Planetary Boundary Layer.
"PB" in Table 2, page 9 is replaced by "PBL (Planetary Boundary Layer)".
The authors wish to thank the anonymous referee for the very helpful comments and corrections. All corrections have been included in this new version. A response to the general and specific comments is provided below (in blue).

**#General comments :**

This paper is about the modeling of wildfires in air quality models and the associated uncertainties on the aerosols concentrations and optical properties. The article is based on several numerical simulations of the 2007 fires in the Mediterranean region with the Polyphemus and CHIMERE models. Simulations explore the sensitivity of these models to chemical related factors (mainly VOC emissions) and dynamics (injection height) . The scientific approach is sound. The work presented is substantial (several simulations have been done and analyzed) but the conclusions and discussions deserve more work before publication. Several conclusions written in the manuscript are expected and already known. The risk is that the impact of the publication to a broader scientific community remains limited unless the authors put the conclusions into a wider perspective. In particular, this would require to add a section dedicated to a scientific discussion including more references to previous works on the subject if possible.

A scientific discussion including  previous work highlighting the impact of fire on the injection height is already mentioned in section 4 page 19 line 24  to page 20 line 7.

To highlight the importance and meaning of the sensitivity studies, the sentences line 29-32 page 24 : "The sensitivity to I/S-VOCs emissions and injection height is important as it shows the maximum impact of I/S-VOCs emissions and injection height on PM2.5 concentrations and AOD. The sensitivity is high (40 to 50%), but it may not explain order of magnitude differences that sometimes occur between models and observations." are replaced by: "Sensitivities to these key parameters are computed using simulations performed with different configurations of Polyphemus. These configurations are chosen to maximize the sensitivities. AOD is particularly sensitive to I/S-VOCs emissions (up to 40% sensitivity, while surface PM2.5 concentrations are particularly sensitive to injection heights (up to 50% sensitivity). These sensitivities are most of the time higher than inter-model sensitivities, which are mostly linked to the model vertical discretization close to fire emissions."

To broaden the conclusion  and address  the limits of our study following comments from another  reviewer, paragraphs on other existing  sources of uncertainties are added in the conclusion:

(1-The influence of NH3 emissions and the formation of inorganics is added in page 27 line 26- page 28 line 4 in the revised paper)
"Over Europe, inorganics (mainly sulfate, sea salt and ammonium) contribute highly to

[revised manuscript text omitted]

**#Specific comments :**

**1) The added value of the CHIMERE model to the analysis must be better explained. The main conclusion I hold is the impact of the vertical resolution. The vertical resolution is certainly a factor of uncertainty for the representation of the boundary layer in general (with or without fire) and would deserve a full sensitivity analysis. At a minimum, this limitation of both models should be discussed.**

A) The added value of the CHIMERE model to the analysis must be better explained:

These sentences are added in page 4, lines 18 – 20 in the revised paper: "CHIMERE simulations (Menut et al., 2013) allow us to perform inter-model comparison and to evaluate the capability of other current CTMs to simulate the impact of wildfires on the regional particulate matter budget and to quantify the uncertainties on air quality modeling related to the integration of fire emissions in CTMs."

B) The main conclusion I hold is the impact of the vertical resolution. The vertical resolution is certainly a factor of uncertainty for the representation of the boundary layer in general (with or without fire) and would deserve a full sensitivity analysis. At a minimum, this limitation of both models should be discussed:

The sentence where we pointed out the impact of the vertical resolution in the inter-model sensitivity (section 4 page 19 lines 14-16) is modified to stress the limitation of both models: "Furthermore, this region of high inter-model sensitivity corresponds to the region where the sensitivity to the injection height is the highest for PM2.5 concentrations. It may therefore be linked to differences in the models' vertical discretization. The models use different vertical coordinates and different numbers of vertical levels. The vertical resolution of the models is rather low as Polyphemus uses 14 vertical levels and CHIMERE uses 19 vertical levels."

**2) The introduction refers to the study of PM2.5 and PM10. The latest are little discussed.**

In this work, we focus only on PM2.5 since PM10 concentrations in the Euro-Mediterranean area are strongly affected by dust, which are difficult to simulate due to their sporadic nature and their main sources are located out of the model domain.

The sentence in the introduction page 3 lines 23-24 "The objective of this study is to evaluate the capabilities of current CTMs to simulate the impact of wildfires on the regional particulate matter budget (PM2.5 and PM10)." is replaced by : " The objective of this study is to evaluate the capabilities of current CTMs to simulate the impact of wildfires on the regional particulate matter budget. In the Mediterranean region, surface PM10 is dominated by the contribution from dust (Rea et al., 2015). Since the focus of this study is on biomass burning, the discussion is centered on the simulation of surface PM2.5. The total loading of aerosols over the region is evaluated using comparisons of AOD to observations."

**3) The fact that only one ground station was available to validate the model near fires moderates confidence in the findings. The paucity of the in-situ data (which is not the fault of the authors) should be pointed out in the general conclusion.**

Sentences in the conclusion page 24 lines 4-7: "The ability of Polyphemus/Polair3D and CHIMERE to simulate regional PM2.5 concentrations and AOD was evaluated based on comparison to available measurements. The general evaluation compared to background AIRBASE and AERONET surface observations shows good performances (with high correlation coefficients) for surface PM2.5 concentrations and AOD." are replaced by:

"The ability of Polyphemus/Polair3D and CHIMERE to simulate regional surface PM2.5 concentrations and aerosol optical depth (AOD) was evaluated based on comparison to available measurements (8 AIRBASE stations and 6 AERONET stations). Only two out of the 8 AIRBASE stations (GR0039A and GR0035A in Greece) and 3 out of 6 AERONET stations (Lecce University in Italy, Blida in Algeria and Bucharest in Romania) are used, because they are the most affected by wildfires (background suburban and rural stations with fire contribution higher than 10%). The lack of surface observations strongly limits this evaluation but it is partly complemented by comparisons to MODIS satellite-based observations of AOD.Comparisons to surface and remote sensing observations show that the models can simulate enhancements of a good order of magnitude and +/- 1 day uncertainty in the timing."

These sentences are added to page 27 lines 9-11: "More surface observations in remote regions would be necessary for a precise evaluation of the simulated long-range transport from fire emissions, the aerosol speciation within the plumes and the resulting impact on air quality."

**4) Both models include wet and dry deposition but little information is provided on the approaches. Are dry and wet depositions sources of uncertainties for PM2.5 and AOD? Wet deposition might not be predominant during the studied period.**

*4.a) Both models include wet and dry deposition but little information is provided on the approaches.*

The sentence in page 5 line 14: "Both models include wet and dry deposition" is replaced by : "Both models include wet and dry deposition. Deposition in Polyphemus is described in Sartelet et al. (2007) and in CHIMERE in Menut et al. (2013) and Mailler et al. (2017) ".

For particles, dry deposition is parameterized with a resistance approach following Zhang et al. (2001). In Polyphemus, the in-cloud and below-cloud scavenging is parameterized following Roselle and Binkowski (1999).

***4.b) Are dry and wet depositions sources of uncertainties for PM2.5 and AOD? Wet deposition might not be predominant during the studied period.***

In this study, we didn't investigate the uncertainties due to dry/wet deposition over the Euro-Mediterranean region. However, Roustan et al. (2010) pointed out the importance of dry and wet deposition over Europe and studied the sensitivity of air quality model (Polyphemus) to input data over the Europe with a focus on aerosols. They found that PM is sensitive to options influencing deposition such as wet diameter and aerosol density. They found during both summer and winter, the uncertainties on wet diameter and aerosol density can reach 19% and 9% respectively. Uncertainties on these parameters can lead to uncertainties on the dry/wet deposition fluxes and thus on AOD and PM2.5 concentrations. A discussion on these uncertainties is added in the conclusion (page 28 lines 16-21 in the revised paper).

**5 ) The conclusion on the three months period evaluation says that the simulations are improved when fire emissions are included: is this surprising? It seems to me that the conclusion was expected unless there are previous studies that have concluded otherwise?**

Indeed, it is not surprising that adding fire emissions, the simulations are improved. But, the main goal of this section is to evaluate statistically the performance of two CTMs to simulate regional PM2.5 concentrations and optical properties when fire emissions are considered, based on comparisons to available measurements. The main conclusion is that models are in good agreement with observations only when considering fire emissions. This shows the importance of considering these emissions during fire events but also gives confidence in the emissions included. Also, it compares the uncertainties linked to modeling the injection height, I/S/L-VOC emissions, and inter-model uncertainty to determine whether it is relevant to keep on improving models of injection height and I/S/L-VOC emissions. Conclusions to this point were strengthen in the current version.

**6) Surface PM2.5 are particularly sensitive to the injection heights" is also an expected conclusion. Can the study help to decide between a PBL mixing of fire emissions vs. injection height above PBL?**

In Table 3 and 4, the model shows almost similar statistics and performance using both configurations,in simulations Poly-ref (injection between 1 km and PBL) and Poly-3km (injection between 1 and 3 km). Therefore this study can not really help to decide between a PBL mixing of fire emissions .vs. injection above PBL. Injection heights should actually vary as a function of meteorological conditions and fire intensity using a pyroconvection scheme. Here we tested two extreme situations. We have already specified that in section 2.3 line 5: "This choice of sensitivity study may be viewed as conservative since, for example, injection heights are limited to 3 km. But it is also extreme since maximum injection at 3 km is imposed to all fires, resulting in injection above the boundary layer. This could be realistic, since based on the Multi-angle Imaging SpectroRadiometer (MISR) observations, however Mims et al. (2010) estimated that 26% of the fire plumes exceed the boundary layer."

**7) The introduction of the PPMfine fraction remains a little obscure for me and would deserve a little more explanation. It is important to understand this variable in light of its significant contribution to the composition of PM2.5 and the uncertainties in its definition (expected overestimation). How is this missing part treated in the other models?**

PPMfine are all the unidentified fine particles emitted by wildfires which are incorporated to consider the differences between PM2.5 emissions and the total of all PM included in specific species. There are a lot of uncertainties related to the estimation of PPMfine emissions.
Several models treat this missing part by considering I/S/L-VOCs emissions from particulate matter emissions from fire emissions (Koo et al., 2014; Konovalov et al., 2015). For example, several studies estimated the I/S/L-VOCs emissions by multiplying the primary organic aerosols (POA) by a factor of 1.5 following the chamber measurements (Robinson et al., 2007; Zhu et al., 2016; Kim et al., 2016).
Moreover, other studies/models do not consider specific species/ surrogates to treat these missing emissions, they use a ratio to reduce uncertainties related to the estimation of PM emissions. For example, Kaiser et al. (2011) use a factor of 3.4 for PM emissions based on the comparison between simulations and AOD observations.

**#Technical corrections:**

**1) Figure 2: I assume that these are emissions calculated by APIFLAMME ?**

The title of Figure 2 in page 8: "Daily total OC emissions during the summer of 2007 in the four sub-regions of Figure1"Is replaced by: " Daily total OC emissions calculated by APIFLAME during the summer of 2007 in the four sub-regions of Figure 1".

**2) Section 2.1: how are I/S/L VOCs represented in CHIMERE?**

In CHIMERE, we do not consider I/S/L-VOCs.

**3) Table 2: I am confused between the Table marks and the comment above concerning the configuration of the CHIMERE model. The comment refers to simulation without I/S-VOCs and with fires but the table for the CHIMERE-ref shows a "Yes" for I/S-VOCs**

In Table 2 page 9:  "yes" for I/S-VOCs from fire for simulation " CHIMERE-ref" is corrected and replaced by "No".

**4) Change PB to PBL.**

In Table 2 page 9 : "Between 1 km and PB" for Fire emissions' injection height is replaced by " Between 1 km and PBL".

**5) Figure 3 & 4: Legend: add "surface PM2.5"**

The legend in Figure 3 in page 12 "Daily mean PM2.5 and AOD at 550 nm from the Poly-ref simulation averaged over the summer of 2007 (the 8 AIRBASE and 6 AERONET stations, used in

this work, are represented here in blue dots) " is replaced by "Daily mean surface PM2.5 and AOD at 550 nm from the Poly-ref simulation averaged over the summer of 2007 (the 8 AIRBASE and 6 AERONET stations, used in this work, are represented here in blue dots)".

The legend in Figure 4 in page 13 "Left panel: relative difference of PM 2.5 concentrations between simulations Poly-ref and Poly-Nofires during the first fire event. Right panel: relative difference of PM2.5 concentrations between simulations Poly-ref and Poly-Nofires during the second fire event." is replace by: "Left panel: relative difference of surface PM 2.5 concentrations between simulations Poly-ref and Poly-Nofires during the first fire event. Right panel: relative difference of surface PM2.5 concentrations between simulations Poly-ref and Poly-Nofires during the second fire event."

---

## Author Response (AR2)

[revised manuscript text omitted]

**Anonymous Referee #2**

The authors wish to thank the anonymous referee for the very helpful comments and corrections. All corrections have been included in this new version. A response to the general and specific comments is provided below (in blue).

**# General comments:**

**B) Here the authors have misunderstood, and their response is not valid. I was talking about contributions of dust and sea-salt to PM2.5, not PM10. As I stated, and as shown in Putaud, the low-diameter (ie 2.5 um!) tail of these large particles can contribute significantly to PM2.5.**

Indeed, the low-diameter tail of coarse-mode dust (and sea-salt) contributes to PM2.5. In this paper, we want to focus on wildfires. We chose not to consider PM10 in this paper in order to reduce the uncertainties related to dust and sea-salt emissions as much as possible. However, dust and sea-salt emissions are accounted for in both model simulations, so that their influence on PM2.5 is also accounted for. To avoid confusion, we rephrased the sentences in page 10 lines 5-7 "PM10 concentrations in the Mediterranean area are strongly affected by dust , which are difficult to simulate due to their sporadic nature and the fact that their main sources are located out of the model domain. In order to evaluate more specifically the uncertainties associated with fine particles (largest contribution from fires), and to minimize the contribution from dust, the analysis focuses on PM2.5 ." are replaced by " PM10 concentrations in the Mediterranean area are strongly affected by dust, which are difficult to simulate due to their sporadic nature and the fact that their main sources are located out of the model domain. Although dust emission is modeled with state-of-the-art parameterization in this study, the analysis focuses on PM2.5 in order to evaluate more specifically the uncertainties associated to wildfires and to minimize the contribution from dust."

**# Other points:**

**3. Again, a misunderstanding. The authors are suggesting that Robinson et al used volatility limits of e.g. 1e4. They didn't, they used a bin centered at 1e4. There is a clear difference between the central value of a bin and its limits, and the limits should be at e.g. 0.32 etc.**

Indeed, Robinson et al. (2007) use values of the saturation concentration bin center and not the volatility limits. Following the reviewer's advice, we modified to text to include the bin limits as stated by Murphy et al. (2014) [Murphy, B. N., Donahue, N. M., Robinson, A. L., and Pandis, S. N.: A naming convention for atmospheric organic aerosol, Atmos. Chem. Phys., 14, 5825-5839, https://doi.org/10.5194/acp-14-5825-2014, 2014.]

The sentences in the paper page 3 lines 7-11 "However, secondary organic aerosols (SOA) are produced through gas-to-particle of oxidation products of volatile organic compounds (VOCs) (with saturation concentration $C^*$ higher than $10^6$ $\mu g.m^{-3}$), intermediate organic compounds (I-VOCs) (with saturation concentration $C^*$ in the range of $10^4 - 10^6$ $\mu g.m^{-3}$) and semi-volatile organic compounds (S-VOCs) (with saturation concentration $C^*$ in the range 0.1–$10^4$ $\mu g.m^{-3}$), Low volatility organic compounds (L-VOCs) (with saturation concentration $C^*$ lower than 0.1 $\mu g.m^{-3}$) " are replaced by : "However, secondary organic aerosols (SOA) are produced through gas-to-particle

of oxidation products of volatile organic compounds (VOCs) (with saturation concentration $C^*$ higher than $3.2 \times 10^6$ µg.m$^{-3}$), intermediate organic compounds (I-VOCs) (with saturation concentration $C^*$ in the range of $320 - 3.2 \times 10^6$ µg.m$^{-3}$) and semi-volatile organic compounds (S-VOCs) (with saturation concentration $C^*$ in the range $0.32$–$320$ µg.m$^{-3}$), Low volatility organic compounds (L-VOCs) (with saturation concentration $C^*$ lower than $0.32$ µg.m$^{-3}$) (Murphy et al. 2014)."

**6. My original comment was that Table 1 was very vague and lacking detail, and it still is. The authors have added more detail to the text, but this would be better in the Table.**

More details (described in the text) are added to Table 1. The clarifications below are added in the revised paper.

| | |
|---|---|
| Meteorology | European Center for Medium-Range Weather Forecasts (ECMWF, ERA-Interim) model |
| Boundary conditions | From nesting simulation: large domain (0.5°x0.5°, horizontal resolution) covering Europe and North Africa (Figure A1 in Appendix A) |
| Chemical mechanism | - Polyphemus: Carbon Bond 05 model (CB05) (Yarwood et al., 2005) for the gas-phase chemistry (modified following Kim et al. (2011) for SOA formation)

- CHIMERE: Modele lagrangien de chimie de l'ozone a l'echelle Regionale 2 (Melchior-2) (Derognat et al., 2003) |
| Horizontal resolution | - Large domain: 0.5°×0.5°
- Small domain: 0.25°×0.25° |
| Vertical resolution | - Polyphemus: 14 levels (surface–12km)
- CHIMERE: 19 levels (surface–200hPa) |
| Biogenic emissions | Model of Emissions of Gases and Aerosols from Nature (MEGAN) (Guenther et al., 2006)
- Polyphemus: the standard MEGAN LHIV database MEGAN-L
- CHIMERE: MEGAN v2.04 |
| Anthropogenic emissions | EMEP emissions inventory for 2007 (European Monitoring and Evaluation Program,www.emep.int) |
| Fire emissions | APIFLAME fire emissions' model v1.0 described in Turquety et al. (2014) |
| Dust emissions | Surface and soil databases (Menut et al., 2013) Briant et al. (2017) |

| Sea-salt emissions | Parameterization of  Monahan (1986) |
|---|---|

**7. The use of odd ratios (e.g. 3:1) for nesting is not restricted to WRF. It ensures that the center of the fine grid matches the center of the coarse grid, and has long been recommended as standard procedure as far as I know. You say that interpolation is done 'properly' in Polyphemus, but provide no evidence for that statement..**

During the simulation of the nesting domain, boundary conditions of the nested domain are computed by interpolating the concentrations of the nesting domain. This interpolation is accurate because it is a 2D interpolation: both domains have the same altitudes, and this altitude does not vary with space and time. Furthermore, both domains use lat-lon coordinates. You can find more details on the numerics of the code in J. Boutahar, S. Lacour, V. Mallet, D. Quélo, Y. Roustan, and B. Sportisse. Development and validation of a fully modular platform for numerical modelling of air pollution: POLAIR. International Journal of Environment and Pollution, 22(1/2):17-28, (2004).

**13. p7,  the authors prefer to retain the notation PPMfine, but I really think this needs to be changed. PPM means primary particulate matter in all publications I am aware of, and there is no need to confuse readers (or referees) by using it for something else. Use REM-PM25 or similar.**

PPMfine is replaced by REM-PM$_{2.5}$ (remaining mass of PM$_{2.5}$) in the new revised version of the paper.

**#Considering the reply to Ref #3 on deposition:**

**Considering the reply to Ref #3 on deposition, you should also address the gas-phase deposition of SVOCs, not just the particle phase. According to Fig.1 of Hallquist et al (ACP, 2009) the vapor-phase deposition is far higher than the particle phase (800 vs 150 TgC/yr).**

The following sentences are added to the conclusion (page 28 lines 21-23 in the new revised version of the paper) to address uncertainties linked to the deposition of SVOCs.

 "Several studies highlights also the importance of the gas-phase deposition of SVOCs (Hodzic et al. 2016; Knote et al., 2015; Karl et al., 2010; Bessagnet et al. 2010; Hallaquist et al., 2009). Hallaquist et al. (2009) highlighted the importance of the vapor deposition which is higher than the particle deposition (800 vs 150 TgC/yr). Knote et al. (2015) showed that the gas-phase SVOCs that are highly water soluble (Henry's law constants H$^*$ between $10^5$ and $10^{10}$M.atm$^{-1}$) are very susceptible to removal processes in the atmosphere (wet deposition and dry deposition). Ignoring the removal of gas-phase SVOCs (dry/wet deposition) in the models can lead to uncertainties on SOA concentrations, AOD and PM2.5 concentrations."

---

## Author Response (AR3)

[revised manuscript text omitted]

**Co-Editor Decision: Publish subject to minor revisions (review by editor) (02 Dec 2018)**
* * *
Dear co-editor,

We wish to thank you again for your corrections.
All corrections have been included in this new version, considering the modifications listed below.
The authors wonder if it is possible to join this paper to ACP ChArMEx special issue.

Best regards,

On behalf of authors
Marwa Majdi
* * *
**1) section 4.1 on Air quality exceedances should be a separate section number 5.**

The section 4.1 on air quality is moved to a separate section (section 5).

**2) most (I would even say all) of the conclusion, as now modified, from page 28 line 28 to the end, expect the last statement, contains mainly a lot of discussion on the uncertainties based on literature. Although this is very useful I consider this should be part of section 4 on uncertainties and not of the conclusions. Therefore I suggest you move it there.**

The text in the conclusion from page 28 line 28 to page 29 line 26 is moved to section 4 as follows:

[revised manuscript text omitted]

**3) please also re-read the conclusions and avoid repetitions with the main text.**

**The conclusions are reworded as follows:**

**The sentences from page 27 line 4 to page 28 line 12 of the old version of the revised paper are replaced by** : "The comparison of surface PM $_{2.5}$ concentrations and  aerosol  optical  depth (AOD) at 550 nm to available surface measurements (background suburban and rural stations only) shows

[revised manuscript text omitted]

---

## Author Response (AR4)

[revised manuscript text omitted]

**Co-Editor Decision: Publish subject to technical corrections (27 Dec 2018)**
* * *
Dear co-editor,

We wish to thank you again for your corrections. All corrections have been included in this new version, considering the modification below.

Best regards, On behalf of authors Marwa Majdi
* * *
**1) I suggest a small addition, page 24, line 1. start the sentence ' Several models...' as follows : 'While a fraction of REM-PM2.5 can be due to the low diameter of dust particles, several models...'**

The sentence in page 24, line 1 "Several models partly treat this missing mass by deducing I/S/L-VOCs contribution from particulate matter emissions (Koo et al., 2014; Konovalov et al., 2015) " is replaced by " While a fraction of REM-PM2.5 can be due to the low diameter of dust particles, several models partly treat this missing mass by deducing I/S/L-VOCs contribution from particulate matter emissions (Koo et al., 2014; Konovalov et al., 2015)".